**Long-term trends of ambient nitrate (NO3-) concentrations across China based on ensemble**
**machine-learning models**
Rui Li[a], Lulu Cui[a] *, Yilong Zhao[a], Wenhui Zhou[a], Hongbo Fu[a,b,c] *
*[a] Shanghai Key Laboratory of Atmospheric Particle Pollution and Prevention, Department of*
*Environmental Science & Engineering, Institute of Atmospheric Sciences, Fudan University,*
*Shanghai, 200433, P.R. China*
*[b] Collaborative Innovation Center of Atmospheric Environment and Equipment Technology*
*(CICAEET), Nanjing University of Information Science and Technology, Nanjing 210044, P.R.*
*China*
*[c] Shanghai Institute of Pollution Control and Ecological Security, Shanghai 200092, P.R. China*
**\* Correspondence to:**
Drs. H. Fu (Email: fuhb@fudan.edu.cn) and L. Cui (Email: 15110740004@fudan.edu.cn)
**Abstract**
High loadings of nitrate ($NO_3^-$) in the aerosol over China significantly exacerbates the air quality
and poses a great threaten on ecosystem safety through dry/wet deposition. Unfortunately, limited
ground-level observation data makes it challenging to fully reflect the spatial pattern of $NO_3^-$ level
across China. Up to date, the long-term monthly $NO_3^-$ datasets at a high resolution were still missing,
which restricted the assessment of human health and ecosystem safety. Therefore, a unique monthly
$NO_3^-$ dataset at 0.25 ° resolution over China during 2005-2015 was developed by assimilating
surface observation, satellite product, meteorological data, land use types and other covariates using
an ensemble model combining random forest (RF), gradient boosting decision tree (GBDT), and
extreme gradient boosting (XGBoost). The new developed product featured excellent cross-
validation $R^2$ value (0.78) and relatively lower root-mean-square error (RMSE: 1.19 μg/m$^3$) and
mean absolute error (MAE: 0.81 μg/m$^3$). Besides, the dataset also exhibited relatively robust





performance at the spatial and temporal scale. Moreover, the dataset displayed good agreement with
($R^2$ = 0.85, RMSE = 0.74 µg/m$^3$, and MAE = 0.55 µg/m$^3$) some unlearning data collected from
previous studies. The spatiotemporal variations of the developed product were also shown. The
estimated $NO_3^-$ concentration showed the highest value in North China Plain (NCP) (3.55 ± 1.25
µg/m$^3$), followed by Yangtze River Delta (YRD (2.56 ± 1.12 µg/m$^3$)), Pearl River Delta (PRD (1.68
± 0.81 µg/m$^3$)), Sichuan Basin (1.53 ± 0.63 µg/m$^3$), and the lowest one in Tibetan Plateau (0.42 ±
0.25 µg/m$^3$). The higher ambient $NO_3^-$ concentrations in NCP, YRD, and PRD were closely linked
to the dense anthropogenic emissions. Apart from the intensive human activities, poor terrain
condition might be a key factor for the serious $NO_3^-$ pollution in Sichuan Basin. The lowest ambient
$NO_3^-$ concentration in Tibetan Plateau was contributed by the scarce anthropogenic emission and
favorable meteorological factors (e.g., high wind speed). In addition, the ambient $NO_3^-$
concentration showed marked increasing tendency of 0.10 µg/m$^3$/year during 2005-2014 ($p$ < 0.05),
while it decreased sharply from 2014 to 2015 at a speed of -0.40 µg/m$^3$/year ($p$ < 0.05). The ambient
$NO_3^-$ levels in Beijing-Tianjin-Hebei (BTH), YRD, and PRD displayed gradual increases at the
speed of 0.13, 0.08, and 0.03 µg/m$^3$/year ($p$ < 0.05) during 2005-2014, respectively. The gradual
increases of $NO_3^-$ concentrations in these regions from 2005 to 2014 were due to that the emission
reduction measures during this period focused on the reduction of $SO_2$ emission rather than $NO_x$
emission and the rapid increase of energy consumption. Afterwards, the government further
strengthened these emission reduction measures, and thus caused the dramatic decreases of $NO_3^-$
concentrations in these regions from 2014 to 2015 ($p$ < 0.05). The long-term $NO_3^-$ dataset over
China could greatly deepen the knowledge about the impacts of emission reduction measures on air



quality improvement. The monthly particulate $NO_3^-$ levels over China during 2005-2015 are open
access in https://doi.org/10.5281/zenodo.3988307 (Li et al., 2020c).
**1. Introduction**
Reactive nitrogen ($N_r$) emissions displayed remarkable increases in the past decades owing to
the high-speed industrial development and urbanization (Cui et al., 2016; Singh et al., 2017).
Ambient reactive N emissions were mainly characterized with nitrogen oxides ($NO_x$), accounting
for about 30% of the gross $N_r$ emissions (Chen et al., 2015; Liu et al., 2011). These important N-
bearing precursors could be transformed into the nitrate ($NO_3^-$) via multiple chemical pathways (e.g.,
heterogeneous or liquid phase reaction), and finally deposited in the terrestrial or aquatic ecosystem
(Jia et al., 2016; Qiao et al., 2015; Zhao et al., 2017). On the one hand, heavy loadings of $NO_3^-$
greatly degraded the atmospheric visibility and cool the surface of the Earth system because
particulate $NO_3^-$ significantly scattered solar radiation (Fu and Chen, 2017). Moreover, enhanced N
deposition might pose a negative effect on the ecosystem health such as biodiversity losses,
freshwater eutrophication, and oceanic acidification (Compton et al., 2011; Erisman et al., 2013).
Hence, deepening the knowledge about the spatial patterns and long-term trends of particulate $NO_3^-$
in the atmosphere is beneficial to accurately evaluate the ecological and environmental effects of N
deposition.
Ground-level observation is often acknowledged to be an effective means to explore the spatial
patterns of ambient $NO_3^-$ concentrations. Many long-term monitoring networks including Clean Air
Status and Trends Network (CASTNET) and Canadian Air and Precipitation Monitoring Network
(CAPMoN) were established to quantify the ambient $NO_3^-$ concentration and inorganic N deposition.
Du et al. (2014) revealed that the $NO_3^-$ deposition showed significant decrease across the United
States during 1985-2012 based on these observation data. To date, most of these observation
networks focused on North America and Europe, whereas few monitoring sites were located on East
Asia especially on China. Fortunately, China has constructed some ground-level observation
networks such as CARE-China Observation Network in recent years. On the basis of these
observation networks, the overall spatiotemporal trend of particulate $NO_3^-$ concentration has been
clarified (Wang et al., 2019c; Xu et al., 2018a). Xu et al. (2018a) observed that the particulate $NO_3^-$
concentration (< 4.5 μm) over China did not show significantly temporal variation during 2011-
2015. Very recently, Wang et al. (2019) found that the $NO_3^-$ level in the fine particle ($PM_{2.5}$)
decreased by 34% during 2015-2017. Although the overall spatial patterns have been preliminarily
revealed based on these isolated sites, these sparse ground-observed sites might not reflect the high-
resolution $NO_3^-$ pollution across China because each station only possessed limited spatial
representative and $NO_3^-$ concentration was often highly variable in space and time (Liu et al., 2017a).
More importantly, the current studies only investigated the ambient $NO_3^-$ concentrations in recent
years, while the long-term variation of $NO_3^-$ level remained unknown. It was well known that the
energy consumption in China displayed remarkable increase in recent decades (Zhan et al., 2018).
Meanwhile, Chinese government also proposed pollutant emission reduction policies since 2005 to
ensure the coordinated development of economic growth and environmental protection (Ma et al.,
2019). However, the synergistic effects of air pollution control policies and increased energy
consumption on long-term evolution trend of $NO_3^-$ pollution over China were not assessed yet,
which were extremely critical for the implementation of emission control measures.
To complement the gaps of ground-level observations, satellite product of $NO_2$ is regarded as a
welcome addition to investigate the long-term trends of N-bearing components in the atmosphere.



Ozone Monitoring Instrument (OMI) was regarded as the typical satellite product applied to
simulate the ambient $NO_3^-$ concentration (Liu et al., 2017b; Vrekoussis et al., 2013). Jia et al. (2016)
firstly used the linear regression method to predict the $NO_3^-$ levels and dry deposition fluxes at the
global scale based on OMI-derived $NO_2$ column amount. However, the dry deposition fluxes of
$NO_3^-$ modelled by Jia et al. (2016) showed weak correlation with the measured value (R = 0.47),
which might be attributable to the simple linear assumption between $NO_2$ column amount and $NO_3^-$
deposition flux. It was well documented that the nonlinearity relationship between multiple
predictors and $NO_3^-$ concentration were hard to reveal on the basis of the simple linear model (Zhan
et al., 2018a; Zhan et al., 2018b). To enhance the predictive performance of $NO_3^-$ concentration, Liu
et al. (2017) used the chemical transport models (CTMs) to estimate the dry deposition fluxes of N-
bearing species recently based on the remotely sensed $NO_2$ column amount. However, CTMs often
suffered from high uncertainty because of the limited knowledge about the generation pathways for
particulate $NO_3^-$ in the atmosphere (Zhan et al., 2018a). Recently, the emergence of machine
learning models provided unprecedented opportunities to estimate the concentrations of N-bearing
components (Chen et al., 2019b; Zhan et al., 2018b). It was well known that the machine learning
models generally showed the better predictive accuracy than CTMs and traditional statistical models
when the training samples were sufficient (Zang et al., 2019; Zhan et al., 2017). In the pioneering
studies, the $NO_2$ estimation has aroused widespread concern (Zhan et al., 2018b; Chen et al., 2019).
Zhan et al. (2018b) employed random forest (RF) coupled with spatiotemporal Kriging model to
simulate the ambient $NO_2$ levels over China, and achieved the moderate modelling performance ($R^2$
= 0.62). Afterwards, Chen et al. (2019) used the extreme gradient boosting (XGBoost) model
combined with kriging-calibrated satellite method to estimate the national $NO_2$ concentration and



significantly improved the predictive performance ($R^2 = 0.85$). Up to date, no study utilized the
machine-learning models to significantly improve the predictive accuracy of $NO_3^-$ concentration.
Moreover, nearly all of the current studies only focused on the spatial pattern of particulate $NO_3^-$
level in China (Liu et al., 2017; Jia et al., 2016), while they cannot establish a long-term $NO_3^-$ dataset
across China.

Here, we firstly developed a high-resolution (0.25°) monthly $NO_3^-$ dataset across China during

2005-2015 based an ensemble model including RF, XGBoost, and gradient boosting decision tree
(GBDT) algorithms. At first, the modelling performance and improvement of this new-developed
product compared with previous datasets were evaluated. Afterwards, we analyzed the spatial
variation and long-term evolution trend of estimated $NO_3^-$ concentration over China and explored
the potential impacts of air pollution control measures on $NO_3^-$ variation. The long-term $NO_3^-$
datasets could supply scientific judge for policy makers to mitigate the severe nitrate pollution in
China.
**2. Input data**
2.1 Ground-level $NO_3^-$ data

The monthly $NO_3^-$ monitoring data during 2010-2015 were collected from NNDMN including

32 sites (Fig. 1 and Fig. S1), and these sites could be divided into three types including urban, rural,
and background sites (Xu et al., 2018a). Ambient concentrations of particulate $NO_3^-$ were
determined on the basis of an active DELTA (DEnuder for Long-Term Atmospheric sampling)
system. The system comprises of a pump, a filter sampling instrument, and a dry gas meter with
high sensitivity. Two set of filters in a 2-stage filter pack was applied to sample the aerosol particles,
with a first $K_2CO_3$/glycerol impregnated filter to obtain $NO_3^-$ particles. All of the monitoring sites



kept the same sampling frequency at the month scale. The detailed sampling and analysis procedures
have been described by Xu et al. (2018). The detection limit of particulate $NO_3^-$ concentration over
China is 0.05 μg/m$^3$.
2.2 Satellite product of $NO_2$ column density

The OMI-$NO_2$ level-3 tropospheric column densities (0.25° resolution) were used to predict the

$NO_3^-$ concentration (Fig. S2). The OMI aboard on the Aura satellite was available since September,
2004, which displayed global coverage and crossed the entire earth each day. OMI possessed three
spectral channels ranging from 270 to 500 nm, and thus was often applied to monitor the gaseous
pollutants such as $NO_2$, $SO_2$, an $O_3$.

In this study, we downloaded the daily $NO_2$ columns during 2005-2015 from

https://earthdata.nasa.gov/. The tropospheric $NO_2$ column density data of poor quality (e.g., cloud
radiance fraction > 0.5, solar zenith angles > 85°, and terrain reflectivity > 30%) should be removed.
Additionally, the cross-track pixels sensitive to significant row anomaly also must be deleted.
Finally, the monthly $NO_2$ columns were estimated by averaging the daily $NO_2$ columns.
2.3 Meteorological factors, land use types, and other variables

These independent variables for particulate $NO_3^-$ estimates were gained from multiple sources.

The meteorological data on a daily basis were downloaded from ERA-Interim datasets (0.25°
resolution) in the website of http://www.ecmwf.int/ (Table S1). Among all of the daily
meteorological data in ECMWF website, 2-m temperature ($T_{2m}$), 2-m dewpoint temperature ($D_{2m}$),
10-m U wind component ($U_{10}$), 10-m V wind component ($V_{10}$), sunshine duration (Sund), surface
pressure (Sp), boundary layer height (BLH), and total precipitation (Tp). The elevation, gross
domestic production (GDP), and population density (PD) data over China were downloaded from



the website of http://www.resdc.cn/. PD and GDP in 1995, 2000, 2005, 2010, and 2015 were linearly
interpolated to calculate PD and GDP in each year. Afterwards, these data were incorporated into
the final model to predict the particulate $NO_3^-$ concentration over China. In addition, the land use
data (e.g., grassland, forest, urban, and agricultural land) were also downloaded from the website of
http://www.resdc.cn/.
These independent variables collected from various sources were uniformly resampled to 0.25°
× 0.25° grids. For instance, the land use area, GDP, and PD in 0.25° grid was calculated based on
area-weighted average algorithm. To ensure the better predictive performance, it was necessary to
employ the appropriate variable selection method to remove some redundant predictors. The basic
principle of the variable choice was to remove the variables with the lower importance values. The
variables could be regarded as the redundant ones when the $R^2$ value of the final model showed
dramatic decrease after removing them.
**3. Methods**
3.1 Ensemble model development
In the previous studies concerning about air pollution prediction, RF, gradient boosting decision
tree (GBDT), and extreme gradient boosting (XGBoost) showed good predictive performance (Li
et al., 2020a). RF model possesses a large amount of decision trees, and each one suffered from an
independent sampling process and these trees displayed the same distribution (Breiman, 2001). This
model generally shows the higher prediction accuracy due to the injected randomness. The model
performance mainly relies on the number of trees, the variable group, and the splitting features. The
detailed algorithms are shown as follows:
$$f(x) = \sum_{z=1}^{Z} c_z I(x \in M_z) \quad (1)$$
$$\overset{\Delta}{c_z} = mean(y_i \mid x_i \in M_z) \quad (2)$$
$$L_1(m,n) = \{X \mid X_j \le n\} \,\&\, L_2(m,n) = \{X \mid X_j > n\} \quad (3)$$
$$\min_{m,n}\left[ \min \sum_{M_1(m,n)} (y - c_1)^2 + \min \sum_{M_2(m,n)} (y - c_2)^2 \right] \quad (4)$$
$$\overset{\Delta}{c_1} = mean(y_i \mid x_i \in M_1(m,n)) \,\&\, \overset{\Delta}{c_2} = mean(y_i \mid x_i \in M_2(m,n)) \quad (5)$$
where $(x_i, y_i)$ denotes the sample for $i = 1, 2, \ldots, N$ in M regions $(M_1, M_2, \ldots, M_z)$, $c_m$ represents
the response to the model, $\overset{\Delta}{c_z}$ denotes the best value, m represents the feature variable, and n is
the split point.
GBDT model is often considered to be a typical boosting method. Compared with RF model,
each classifier is applied to decrease the residual of the last round. The detailed equations are as
follows:
$$c_{tj} = \arg\min \sum_{xi \in Rt_j} L(y_i, f_{t-1}(x_i) + c) \quad (6)$$
$$f_t(x) = f_{t-1}(x) + \sum_{j=1}^{J} c_{tj} I \quad (7)$$
$c_{tj}$ denotes the predicted the estimation error in the last round; yi represents the observed value;
$f_{t-1}(x_i)$ is the predicted value in the last round. c was regarded as the optimal value when $c_{tj}$ reaches
the least value.
XGBoost method is an updated version of GBDT model and loss functions are expanded to the
second order function. On the basis of the pioneering studies (Chen et al., 2019a), XGBoost
generally shows excellent performance because of its high efficiency and impressive accuracy. The
detailed XGBoost algorithm is shown as the following formula (Zhai and Chen, 2018):
$$L^{(t)} = \sum_{i=1}^{n}[l(y_i, \overset{\wedge}{y}^{(t-1)}) + \partial_{y^{(t-1)}} l(y_i, \overset{\wedge}{y}^{(t-1)}) f_t(x_i) + \frac{1}{2}\partial^2_{y^{(t-1)}} l(y_i, \overset{\wedge}{y}^{(t-1)}) f_t^2(x_i)] + \Omega(f_t) \quad (8)$$



198 where L$^{(t)}$ represents the cost function at the t-th period. $l$ is the differentiable convex loss function

199 that reveals the difference of the predicted value ( $\overset{\wedge}{y}$ ) of the i-th instance at the t-th period and the

200 target value (y$_i$). f$_t$(x) denotes the increment.

201  However, each model still shows some disadvantages in the prediction accuracy. Consequently,

202 it was proposed to combine these models with multiple linear regression (MLR) model to further

203 estimate monthly NO$_3^-$ concentration in the atmosphere over China. As shown in Fig. 2, three

204 submodels including RF, GBDT, and XGBoost were stacked through MLR model to estimate the

205 monthly NO$_3^-$ concentration over China. At first, a 5-fold cross-validation method was adopted to

206 train each submodel to determine the appropriate parameter. Afterwards, the MLR model was

207 trained with the final simulated concentrations of three submodels and observations. Finally, the

208 high-resolution ambient NO$_3^-$ level over China were estimated based on the optimal ensemble model.

209 The detailed algorithms are shown as follows (Fig. 2):

210    $NO_3^- = \text{A} \times \text{Pred\_RF} + \text{B} \times \text{Pred\_GBDT} + \text{C} \times \text{Pred\_XGBoost} + \text{e}_{ij}$ (9)

211 where Pred_RF, Pred_GBDT, and Pred_XGBoost denote the predicted NO$_3^-$ concentrations by RF,

212 GBDT, and XGBoost, respectively. A, B, and C represent the partial regression coefficients of RF,

213 GBDT, and XGBoost predictors, respectively.

214  The RF model was trained using matlab2019a with a package named random forest-master. Both

215 of GBDT and XGBoost algorithms were conducted using many packages named *gbm*, *caret*, and

216 *xgboost* in R software.

217 3.2 The error estimation and uncertainty assessment

218  The estimation performance of the ensemble model was evaluated based on 10-fold cross-

219 validation algorithm. The principle of this method meant that the entire datasets were divided into



10 groups with the same capacity randomly. Nine groups were applied to develop the model and the
remained one was used to predict the $NO_3^-$ level. After ten rounds, every observed $NO_3^-$
concentration showed a corresponding predicted value. Some key indices such as determination
coefficient ($R^2$), root mean square error (RMSE), and mean absolute prediction error (MAE) were
selected as the key indicators to identify the optimal modelling method.

The uncertainty of ensemble model were mainly derived from input ancillary variables. For

instance, both of the satellite data and meteorological data often suffered from some uncertainties.
To quantify the uncertainties derived from meteorological data, the meteorological data at 0.25°
across China were validated using ground-measured meteorological data downloaded from the
website of Chinese Meteorology Bureau (http://data.cma.cn/). Additionally, $NO_2$ columns generally
suffered from some uncertainties, whereas the uncertainties of these $NO_2$ columns cannot be
determined because the data about the ground-level $NO_2$ columns were not open access. In our study,
we only estimated the missing ratio of $NO_2$ column, thereby evaluating the uncertainty of $NO_3^-$
dataset.
3.3 Trend analysis

The trend analysis of particulate $NO_3^-$ concentration was performed using the Mann-Kendall

nonparametric test. This method has been widely applied to analyze the historical trends of carbon
fluxes (Tang et al., 2019) and air quality (Kong et al., 2020), which could reflect whether these data
suffered from significant changes at a significance level of 0.05.
**4.   Results and discussion**
4.1 Descriptive statistics of observed $NO_3^-$ concentrations

The ensemble model were applied to fit the $NO_3^-$ estimation model based on 1636 matched



samples across China during 2010-2015. In general, the site-based $NO_3^-$ concentration over China
ranged from 0.3 $\mu g/m^3$ in Bayinbrook of Xinjiang province to 7.1 $\mu g/m^3$ in Zhengzhou of Henan
province with the mean value of 2.7 ± 1.7 $\mu g/m^3$. The monthly $NO_3^-$ concentrations displayed the
highest and lowest values in North China Plain (NCP) and Tibetan Plateau, respectively. Besides,
the monthly $NO_3^-$ level exhibited significantly temporal variation during 2010-2015. The ambient
$NO_3^-$ concentrations in most of sites displayed the gradual increase during 2010-2014, while they
decreased sharply from 2014 to 2015. The spatiotemporal variation of ambient $NO_3^-$ concentration
over China shared similar characteristic with $NO_2$ column amount and urban land area (Fig. S2).
The Pearson correlation analysis revealed that the monthly $NO_3^-$ level showed the significantly
positive relationship with $NO_2$ column amount ($r = 0.57$, $p < 0.01$) and urban land area ($r = 0.35$, p
$< 0.05$) (Fig. S3). However, $D_{2m}$ showed the remarkably negative correlation with ambient $NO_3^-$
concentration ($r = -0.31$, $p < 0.05$).
4.2  The validation of new-developed $NO_3^-$ dataset and comparison with previous products

In our study, the ensemble model was applied to develop a monthly $NO_3^-$ dataset over China

based on various predictors. Besides, other three individual models were also trained to compare
with their predictive performances. The cross-validation result indicated that the $R^2$ value of the new
product developed by ensemble decision trees model reached 0.78, significantly higher than those
developed by RF (0.57), GBDT (0.73), and XGBoost (0.45). Nonetheless, both of RMSE and MAE
exhibited the opposite trends. The RMSE value was in the order of XGBoost (1.98 $\mu g/m^3$) > RF
(1.67 $\mu g/m^3$) > GBDT (1.35 $\mu g/m^3$) > ensemble model (1.19 $\mu g/m^3$). The MAE value followed the
similar characteristic with the order of XGBoost (1.29 $\mu g/m^3$) > RF (0.99 $\mu g/m^3$) > GBDT (0.95
$\mu g/m^3$) > ensemble model (0.81 $\mu g/m^3$). Wolpert (1992) suggested the combination of various
machine-learning models can significantly strengthen the transferability of models. Chen et al.
(2019a) demonstrated that the ensemble model significantly outperformed the individual machine-
learning model because the ensemble model can overcome the weaknesses of individual model.
Besides, we also assessed the annual modelling performance of $NO_3^-$ estimation. Figure S4 shows
that the $R^2$ value of annual $NO_3^-$ estimation reached 0.81, slightly higher than monthly $NO_3^-$
prediction (0.78). Furthermore, both of RMSE (1.23 $\mu g/m^3$) and MAE (0.85 $\mu g/m^3$) for annual $NO_3^-$
estimation were slightly higher than those of monthly $NO_3^-$ prediction.

The new developed $NO_3^-$ dataset showed the markedly temporal discrepancy. The $R^2$ values of

$NO_3^-$ estimates during 2011-2015 (0.88, 0.89, 0.83, 0.74, and 0.78) were notably higher than that
during 2010 (0.62) (Table 1 and Fig. 3). The relatively lower $R^2$ value in 2010 attested to the
dominant role of sampling size on the predictive accuracy for machine-learning models. The training
samples in 2010 (135 samples) were notably less than those in other years due to the lack of
observation data in spring. However, both of RMSE and MAE were not sensitive to the sampling
size. The higher RMSE and MAE focused on the 2010, 2014, and 2015. The higher RMSE and
MAE observed in 2010 might be contributed by the poor predictive performance, while the higher
RMSE and MAE likely attained to the higher $NO_3^-$ levels during other years. In addition, the
performance of the $NO_3^-$ dataset varied greatly at the seasonal scale. The $R^2$ value was in the order
of summer (0.85) > spring (0.80) = autumn (0.80) > winter (0.75) across China (Table 2). The
seasonal variation of $NO_3^-$ concentration was in contrast to the results of fine particle modelled by
previous studies (Li et al., 2020a; Qin et al., 2018). It was supposed that AOD was sensitive to the
precipitation and relative humidity, and thus showed the worse performance in summer. However,
the predictive accuracy of $NO_3^-$ estimation based on $NO_2$ column amount was closely linked with



the chemical transformation from $NO_2$ to $NO_3^-$.
The $NO_3^-$ dataset also displayed markedly spatial variation. The highest $R^2$ value was observed
in NCP (0.70), followed by Southwest China (0.60), Southeast China (0.59), Northwest China (0.55),
and the lowest one in Northeast China (0.44) (Table 3). The highest $R^2$ value occurring in NCP was
mainly attributable to the largest training samples (> 400) compared with other regions. Southeast
China and Southwest China showed satisfactory cross-validation $R^2$ values because the valid
training samples in both of these regions were higher than 300. Although both of Northeast China
and Northwest China possessed limited training samples (< 200), the predictive performances of
these regions showed significant discrepancy. It was assumed that the sampling sites in Northeast
China were very centralized, while the sampling sites in Northwest China were uniformly
distributed across the whole region. Geng et al. (2018) revealed that the modelling accuracy based
on statistical models were significantly affected by the distribution characteristics of sampling sites.
However, both of RMSE and MAE showed different spatial distributions with the $R^2$ value and
slope of fitting curve. Note that the higher values of RMSE and MAE were concentrated on
Southwest China (2.08 and 1.41 $\mu g/m^3$) and Northwest China (2.06 and 1.38 $\mu g/m^3$) rather than
NCP (1.74 and 1.06 $\mu g/m^3$). There are two reasons responsible for the result. At first, the predictive
performances of Southwest China and Northwest China were significantly worse than that of NCP.
Generally, the poor predictive accuracy meant the higher RMSE and MAE when the absolute
concentrations of $NO_3^-$ for training samples were approximately equal. Moreover, most of the
sampling sites in Southwest China were focused on Sichuan Basin, which often showed severe $NO_3^-$
pollution all the year round. Meanwhile, the annual mean $NO_3^-$ concentrations in Yangling and
Wuwei reached 4.1 and 4.5 $\mu g/m^3$, respectively. The higher loadings of $NO_3^-$ concentrations for

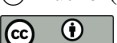

training samples led to the higher RMSE and MAE for Northwest China.
Although the cross-validation result suggested the new developed dataset achieved the better
modelling accuracy, the cross-validation algorithm cannot test the transferability and agreement of
this dataset in the past years. Hence, the unlearning data (annual mean $NO_3^-$ concentration in 10
cities) collected from previous references were employed to validate the transferability of this
product. As shown in Fig. 4, we found that the $R^2$ value of new-developed $NO_3^-$ product and
historical data reached 0.85 (Fig. 4), and the out-of-range $R^2$ value was even slightly higher than the
cross-validation $R^2$ value. Moreover, the out-of-bag slope based on these unlearning data reached
0.81, and equaled to the slope of cross-validation database. The result suggested the new-developed
dataset showed excellent performance in the past decade.
Owing to the severe air pollution issue frequently observed in recent years, especially nitrogen-
bearing haze events, many studies have tried to predict the $NO_3^-$ concentrations in China. Most of
these studies employed CTMs to simulate the ambient $NO_3^-$ concentrations over China. Huang et al.
(2015) employed WRF-CMAQ to estimate the inorganic nitrogen deposition over PRD, and
confirmed that the R value only reached 0.54. Afterwards, Han et al. (2017) used RAMS-GMAQ to
predict the dry deposition flux of reactive nitrogen, and significantly underestimated the $NO_3^-$
concentration in the atmosphere. Very recently, Geng et al. (2019) used CMAQ to estimate the $NO_3^-$
concentrations over East China, and the predictive performance (R = 0.53) showed the similar result
to Huang et al. (2015). Apart from these CTMs, the statistical models also has been applied to
estimate the ambient $NO_3^-$ concentration over China. Unfortunately, the predictive accuracy was not
good based on traditional statistical models (e.g., linear regression) (R = 0.47) (Jia et al., 2016). In
terms of model performance, the developed $NO_3^-$ product in our study was much better than those

330 developed by pioneering studies. Furthermore, this product showed many extra advantages than

331 those obtained by CTMs especially for the hindcast of air pollutants. For instance, CTMs generally

332 required continuous emission inventory data, which were often not available and showed high

333 uncertainties. Moreover, CTMs generally needed substantial computing time and big-data input data

334 to ensure the reliable predictive accuracy. Thus, the $NO_3^-$ product retrieved by CTMs often lacks of

335 long-term dataset (> 10 yr), and our study fills the gaps of previous studies.

336 4.3 Spatial pattern of new-developed $NO_3^-$ dataset

337  The monthly $NO_3^-$ concentration displayed the similar distribution characteristic with $PM_{2.5}$ and

338 $PM_1$ (Wei et al., 2019). Overall, the $NO_3^-$ concentration in East China was much higher than that in

339 West China. The higher $NO_3^-$ concentration was concentrated on NCP ($3.55 \pm 1.25$ μg/m$^3$), followed

340 by Yangtze River Delta (YRD ($2.56 \pm 1.12$ μg/m$^3$)), Pearl River Delta (PRD ($1.68 \pm 0.81$ μg/m$^3$)),

341 Sichuan Basin ($1.53 \pm 0.63$ μg/m$^3$), and the lowest one observed in Tibetan Plateau ($0.42 \pm 0.25$

342 μg/m$^3$) (Fig. 5). Most provinces over NCP such as Beijing, Hebei, Henan, and Shandong suffered

343 from severe $NO_3^-$ pollution due to dense human activities and strong industry foundation (Li et al.,

344 2017), which released a large amount of N-bearing gaseous pollutants to the atmosphere especially

345 in winter. In BTH ($2.97 \pm 1.97$ μg/m$^3$), Wang et al. (2016) verified that these fresh $NO_x$ emitted from

346 power plants or cement industries could be transformed into the nitrate in the particulate phase by

347 the aid of low air temperature. In YRD and PRD, the combustion of fossil fuels and traffic emissions

348 were considered to be the major source of $NO_x$ emission, which favored to the formation of nitrate

349 event through the gas-particle conversion processes (Fu et al., 2017; Kong et al., 2020; Ming et al.,

350 2017). Apart from the contributions of smelting industries, the poor topographical or meteorological

351 conditions were also responsible for the severe $NO_3^-$ pollution in Sichuan Basin (Tian et al., 2017;

Wang et al., 2017). Tibetan Plateau generally showed the clean air quality due to the unique landform
and scarce industrial activity (Yang et al., 2018). In addition, it was interesting to note that the Altai
region and Taklimakan desert in Xinjiang autonomous region also showed some $NO_3^-$ hotspots,
though these regions were often believed to be the remote region. It was assumed that the many
petrochemical industries (e.g., Karamai oil field) were located in the Altai region (Liu et al., 2018).
Besides, Qi et al. (2018) verified that the resuspension of soil dust might trigger the accumulation
of $NO_3^-$ concentration in the aerosol.
4.4 Long-term trend of ambient $NO_3^-$ across China

The temporal variation of $NO_3^-$ levels from 2005 to 2015 over China has been clarified in Fig.

6, Fig. 7 and Table S2. Overall, the ambient $NO_3^-$ concentration in China showed the significant
increasing trend of 0.10 μg/m³/year during 2005-2014, while it decreased sharply from 2014 to 2015
by the speed of -0.40 μg/m³/year. Overall, more than 90% areas of Mainland China showed
consistent temporal variation with the gradual increase from 2005 to 2013, and then rapid decrease
from 2013/2014 to 2015. However, the decreasing/increasing speed displayed significantly spatial
difference in some major regions of China. For instance, the ambient $NO_3^-$ level in BTH showed the
remarkable increase during 2005-2014 by the speed of 0.13 μg/m³/year. Afterwards, the $NO_3^-$ level
decreased rapidly from 2014 to 2015 at a speed of -0.76 μg/m³/year. The $NO_3^-$ concentrations in
YRD (0.08 μg/m³/year) and PRD (0.05 μg/m³/year) both showed the slight increases during 2005-
2014, though the statistical test revealed the increases were significant ($p < 0.05$). However, the
$NO_3^-$ concentrations in YRD and PRD showed the dramatic decreases with -0.79 and -0.59
μg/m³/year, respectively. As seen from 2005 to 2015, the $NO_3^-$ concentration in BTH displayed the
slight increase during this period. Nevertheless, the $NO_3^-$ levels in YRD and PRD both displayed
the slow decreases by the speed of -0.01 and -0.03 µg/m$^3$/year, respectively.

Furthermore, the different provinces displayed disparate temporal variations especially during

11th five year plan (2005-2010). 31 provinces (municipalities/autonomous region) of China can be
classified into three clusters based on the temporal trends of $NO_3^-$ concentrations during 11th five
year plan. The first cluster featured the gradual increase of $NO_3^-$ concentration during this period,
which consisted of three provinces in Northeast China (e.g., Heilongjiang) and central provinces in
South China (e.g., Jiangxi, Anhui) (Table S2). The second cluster represented the provinces with the
stable increases of $NO_3^-$ during 2005-2007 and slight decreases during 2007-2010. Some provinces
of NCP (e.g., Beijing, Hebei, Henan) and Northwest China (e.g., Gansu, Inner Mongolia, Ningxia)
fell into the second cluster. The last cluster featured the opposite temporal trend to the second cluster
during 2005-2010, which included many southern provinces such as Fujian, Guangdong, Zhejiang,
and Guangxi. Although the central government proposed the emission reduction goal in 2006, the
ambient $NO_3^-$ concentrations in most provinces did not display pronounced decreases, which was
totally different from the decrease of $PM_{2.5}$ since 2007 (Xue et al., 2019). Especially in the provinces
of Northeast China (e.g., Liaoning), the ambient $NO_3^-$ concentrations in these provinces still showed
the rapid increases after the proposal of emission control measures. It was assumed that these
provinces generally possessed a large amount of energy-intensive industries and coal-fired power
plants (Zhang et al., 2018). Moreover, the result might be associated with the fact that the emission
reduction measures focused on the reduction of $SO_2$ emission rather than $NO_x$ emission (Kanada et
al., 2013). Schreifels et al. (2012) revealed that major control measures during this period included
shutting down inefficient industries, increasing the pollution levy for excessive $SO_2$ emissions, and
implementing energy conservation projects. Therefore, the total $SO_2$ emission in 2010 decreased by



more than 14% compared with the emission in 1995 and the ambient $SO_2$ concentrations in many
provinces since 2005 displayed significant decreases compared with those in 1990s (Li et al., 2020b;
Lu et al., 2013; Zhou et al., 2015). Nonetheless, the $NO_x$ emission in China did not display
significant decrease during this period (Duncan et al., 2016; Granier et al., 2017), and thus the
ambient $NO_3^-$ in many provinces still kept the higher concentrations. It should be noted that the
$NO_3^-$ concentrations in some provinces of NCP exactly exhibited the slow decreases after 2007. It
was supposed that the energy structure adjustment and elimination of backward production capacity
promoted the small decrease of $NO_3^-$ concentrations (Ma et al., 2019). Unfortunately, the slight
decreases were quickly offset by the rapid increase of energy consumption. Zhang et al. (2018)
demonstrated that the industry added values and private car number in BTH have been increasing
by 189.4% and 279.6% during 2005-2010, respectively.

Since 2010, the central government began to implement severe limitations in $PM_{2.5}$, $NO_x$, and

soot emissions, and thus the total $NO_x$ emission during 11th five year plan (2011-2015) showed
slow decrease (10%) across China (Ma et al., 2019). However, the $NO_3^-$ concentrations across China
did not show rapid response to the emission control measures. For instance, the $NO_3^-$ concentrations
in most provinces of China still showed rapid increases during 2010-2013 (2014) (Fig. 7 and Fig.
8). The result suggested that the control measures about the $NO_x$ emissions from vehicles and ships
might be not very effective. Until 2013, the central government issued Action Plan for Air Pollution
Prevention and Control (APPC-AP) in order to enhance the air pollution prevention measures (Li et
al., 2017; Li et al., 2019). Many powerful economic and policy means including pricing (tax) policy
and optimization of industrial layout caused the rapid decreases of $NO_3^-$ concentrations after 2013
in many provinces (e.g., Beijing, Hebei, Zhejiang). Wang et al. (2019b) also verified that the $NO_3^-$
level in $PM_{2.5}$ over BTH has decreased by 20% during 2013-2015, which was in accordance with
the finding of our study. In addition to the impact of emission reduction, the rapid decrease of $NO_3^-$
concentration over China after 2013 might be linked with the beneficial meteorological factors
because Chen et al. (2019c) has demonstrated that favorable meteorological conditions led to about
20 % of the $PM_{2.5}$ decrease in BTH during 2014-2015. However, the decreasing trend of $NO_3^-$
concentration during 2014-2015 in PRD (-0.59 μg/m³/year) was significantly slower than that in
BTH (-0.76 μg/m³/year) and YRD (-0.79 μg/m³/year) (Table 4). Wang et al. (2019b) found that the
ambient $NO_3^-$ concentration in a background site of PRD even showed an upward trend during 2014-
2016. Thus, it was necessary to strengthen the control of nitrogen oxide emissions.

In general, the ambient $NO_3^-$ concentration varied greatly at the seasonal scale (Fig. 9). China

undergone the most serious $NO_3^-$ pollution in winter (1.57 ± 0.63 μg/m³), followed by autumn (1.09
± 0.52 μg/m³), spring (0.78 ± 0.50 μg/m³), and the lowest one in summer (0.63 ± 0.40 μg/m³) (Table
S3). The higher $NO_3^-$ concentration observed in winter might be contributed by the dense coal
combustion in North China and unfavorable meteorological conditions (Itahashi et al., 2017; Quan
et al., 2014; Wang et al., 2019d). The lightest $NO_3^-$ pollution in summer was attributable to the
abundant precipitation, which promoted the diffusion and removal of pollutants and reduced
ambient $NO_3^-$ level (Hu et al., 2005). The ratio of $NO_3^-$ concentration in winter ($NO_3^-_{winter}$) and that
in summer ($NO_3^-_{summer}$) varied greatly at the spatial scale. The $NO_3^-_{winter}$/ $NO_3^-_{summer}$ in some
provinces (municipalities) including Tianjin (2.11), Hebei (2.25), and Henan (2.84) displayed the
higher values compared with other provinces. The higher $NO_3^-_{winter}$/ $NO_3^-_{summer}$ in NCP might be
affected by the fossil fuel combustion for domestic heating, while some southern provinces did not
need domestic heating in winter. In contrast, the ratio of $NO_3^-_{winter}$/ $NO_3^-_{summer}$ exhibited the lower





values in some western provinces such as Tibet and Qinghai. It might be probabaly associated with
the less aerosol emission from anthropogenic source and the higher wind speed (Wei et al., 2019).
4.5 Uncertainty analysis of $NO_3^-$ estimation
The ensemble model of three machine-learning algorithms captured the better accuracy in
predicting the $NO_3^-$ level from OMI data. Nonetheless, the ensemble model still showed some
improvement space in terms of the $R^2$ value. At first, meteorological data collected from reanalysis
in ECMWF website generally showed high uncertainty, which inevitably increased the error of $NO_3^-$
estimation. In our study, we validated the gridded $T_{2m}$ and Tp datasets against the groud-observed
datasets and found that the $R^2$ values of $T_{2m}$ and Tp reached 0.98 and 0.83 (Table S4), respectively.
The result suggested that $T_{2m}$ showed the lower uncertainty, while Tp displayed relatively higher
uncertainty. Except $T_{2m}$ and Tp, the ground-level datasets for other meteorological factors were not
open access, and thus we cannot assess their uncertainties. Thus, we only reviewed some references
and evaluated their uncertainties. For instance, Guo et al. 2019 found that the reanalysis BLH data
also exhibited large uncertainties because few sounding data were assimilated. These uncertainties
derived from predictors could be passed to the ensemble model, and thus increased the uncertainties
of ambient $NO_3^-$ estimates.
The second reason was closely linked to the missing $NO_2$ column amount across China. The
$NO_2$ column amount retrieval showed many nonrandom biases especially for the arid or semi-arid
area with high surface reflectance. The missing $NO_2$ column amounts over China were not filled in
our study due to the increased uncertainty of filling $NO_2$ column. Moreover, it should be noted that
the monthly $NO_2$ column amounts were averaged based on the daily one, and the missing ratio of
daily $NO_2$ columns during 2005-2015 reached 57.64%, the higher missing ratio might increase the

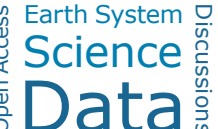

uncertainty of $NO_3^-$ simulation.
Lastly, the developed ensemble model did not integrate the direct spatiotemporal weight
indicators (e.g., the distance of observed sites and contiguous grids) though many predictors (e.g.,
month of year) reflecting spatiotemporal autocorrelation were input into the original model as the
key predictors. Furthermore, the developed model was the ensemble one of three original models,
which ignored the spatiotemporal autocorrelation of estimation residues from first-stage model. In
the future work, the ensemble model could be combined with a space-time model to further enhance
the modelling performance.
**5.   Data availability**
The monthly $NO_3^-$ datasets at 0.25° resolution across China during 2005-2015 are available at
https://doi.org/10.5281/zenodo.3988307 (Li et al., 2020), which can be downloaded in xlsx format.
The missing values are shown in NaN.
**6.   Conclusions and implications**
In this study, RF, GBDT, and XGBoost algorithms were combined to establish a high-resolution
(0.25 °) $NO_3^-$ dataset over China during 2005-2015 on the basis of multi-source predictors. The $NO_3^-$
product showed high cross-validation $R^2$ value (0.78), but low RMSE (1.19 $\mu g/m^3$) and MAE (0.81
$\mu g/m^3$). The $NO_3^-$ dataset showed the markedly spatiotemporal discrepancy. The $R^2$ value was in the
order of summer (0.85) > spring (0.80) = autumn (0.80) > winter (0.75) across China, and the $R^2$
showed the highest value in NCP. In addition, the dataset exhibited excellent transferability ($R^2$ =
0.85, RMSE = 0.74 $\mu g/m^3$, and MAE = 0.55 $\mu g/m^3$) on the basis of the unlearning observed data in
ten sites.
The new-developed $NO_3^-$ dataset showed remarkably predictive accuracy compared with





previous products developed by CTMs and linear regression model. The result might be linked to
two key reasons. First of all, the new product assimilated high-resolution $NO_2$ column amount
instead of the $NO_x$ emission inventory used by CTMs. The imperfect knowledge about the chemical
modules with regard of the $NO_3^-$ formation and the inaccurate emission inventory decreased the
predictive performance of CTMs. In contrast, the new product was obtained using ensemble
machine-learning model, which did not need to consider the photochemical or aqueous process from
gaseous $NO_2$ to particulate $NO_3^-$. Compared with the $NO_3^-$ product estimated by linear regression
model ($R^2 = 0.21$), the new product significantly elevated the modelling performance of $NO_3^-$
concentration. It was supposed that the ensemble model for the development of the new $NO_3^-$ dataset
did not predefine the potential relationships between explanatory variables and $NO_3^-$ level as the
multiple regression model, which must assume the linear linkage between dependent variable and
predictors before model establishment.

On the basis of the such dataset, the spatiotemporal variation of $NO_3^-$ concentration over China

during 2005-2015 were clarified. The annual mean $NO_3^-$ concentration followed the order of NCP
($3.55 \pm 1.25$ μg/m$^3$) > YRD ($2.56 \pm 1.12$ μg/m$^3$) > PRD ($1.68 \pm 0.81$ μg/m$^3$) > Sichuan Basin (1.53
$\pm 0.63$ μg/m$^3$) > Tibetan Plateau ($0.42 \pm 0.25$ μg/m$^3$). The higher $NO_3^-$ concentrations in NCP, YRD,
and PRD were mainly contributed by the intensive industrial and traffic emissions. Sichuan Basin
suffered serious $NO_3^-$ pollution due to the high loadings of aerosols and unfavorable terrain
condition. Tibetan Plateau shared with the lightest $NO_3^-$ pollution because of the scarce
anthropogenic emissions and favorable meteorological factors. Additionally, we also found that the
ambient $NO_3^-$ concentration showed significant increasing trend of 0.10 μg/m$^3$/year during 2005-
2014, while it decreased sharply from 2014 to 2015 at a speed of -0.40 μg/m$^3$/year. The ambient




$NO_3^-$ levels in BTH, YRD, and PRD displayed slight increases at the speed of 0.13, 0.08, and 0.03
$\mu g/m^3$/year, respectively. Afterwards, the $NO_3^-$ concentrations decreased sharply at the speed of -
0.76, -0.79, and -0.59 $\mu g/m^3$/year. Although National Economic and Social Development of China
has issued the emission reduction goal in 2006, the $NO_3^-$ concentrations in most provinces did not
show the significant decreases during 2005-2010. It might be contributed by the increase of energy
consumption and non-targeted emission control measures. Since 2010, the government began to
decrease the $NO_x$ emission over China, whereas the $NO_3^-$ concentrations in many provinces still
showed slight increases during 2010-2014 because the benefits of control measures for $NO_x$
emission could be neutralized by elevated energy consumption along with the rapid economic
development. After 2014, Chinese government issued APPC-AP and further enhanced the emission
control measures, and triggered the dramatic decrease of $NO_3^-$ concentration over China. Apart from
the effect of emission reduction, the favorable meteorological conditions might lead to the rapid
decrease of $NO_3^-$ level over China during 2014-2015. Compared with the powerful emission control
measures, meteorological factors only contributed a small portion of $NO_3^-$ reduction in China.
Besides, the decrease speed of $NO_3^-$ level in China also displayed pronounced spatial heterogeneity
and some background region even featured the upward of air pollutant in recent years. Therefore, it
is still imperative to strengthen the emission reduction measures.

It must be acknowledged that our study still suffers from some limitations. First of all, the $NO_3^-$

dataset was developed by machine-learning models, which lacked of the chemical module
concerning about the transformation pathway from $NO_2$ to $NO_3^-$, and might underestimate the
ambient $NO_3^-$ concentration across China. In the future work, the output results of CTMs including
conversion ratio from $NO_2$ to $NO_3^-$, dry/wet deposition flux of $NO_2$ and $NO_3^-$ in the atmosphere



should be incorporated into the machine-learning model to develop next-generation $NO_3^-$ product.
Second, the low time-resolution (monthly) observation data hindered the daily estimation of $NO_3^-$
concentration. The daily $NO_3^-$ datasets are warranted in the future because it could be used to assess
the potential impact on human health. Besides, the ultrahigh-resolution satellite (TROPOMI) can
allow continuation and enhancement of the spatiotemporal $NO_3^-$ estimation though the OMI product
could capture enough spatial variations across China.
**Acknowledgements**
This work was funded by Chinese Postdoctoral Science Foundation (2020M680589) and National
Natural Science Foundation of China (Nos. 21777025).
**Author contributions**
Rui Li, Lulu Cui, and Hongbo Fu conceived and designed the study. Rui Li, Lulu Cui, Yilong Zhao,
Wenhui Zhou collected and processed the data. Rui Li wrote this paper with contributions from all
of the coauthors.



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

**Fig. 1** Spatial distributions of ground-level NO$_3^-$ monitoring sites used for model establishment. Red
circles represent the ground-level sites during 2010-2015. The colormap denotes the elevation
distribution across China.

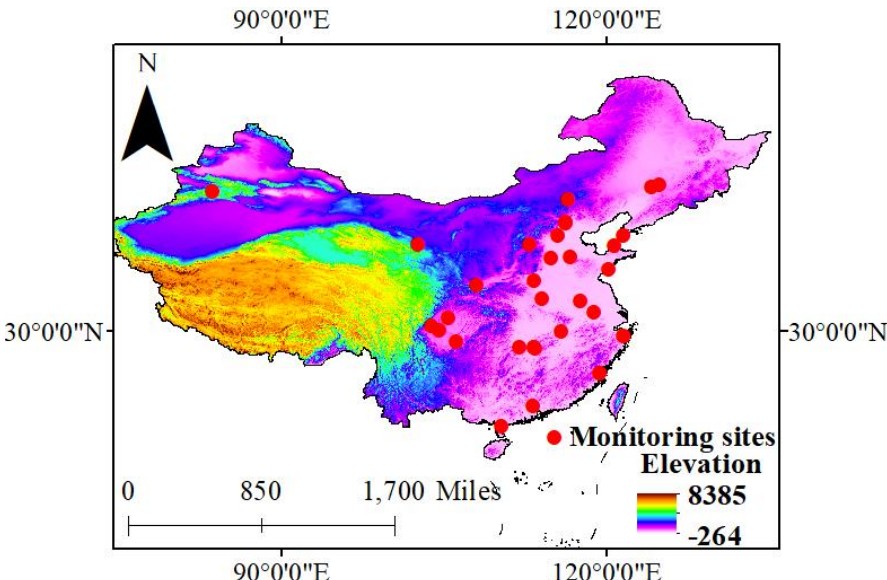




**Fig. 2** The workflow of the ensemble model development for ambient $NO_3^-$ estimates.

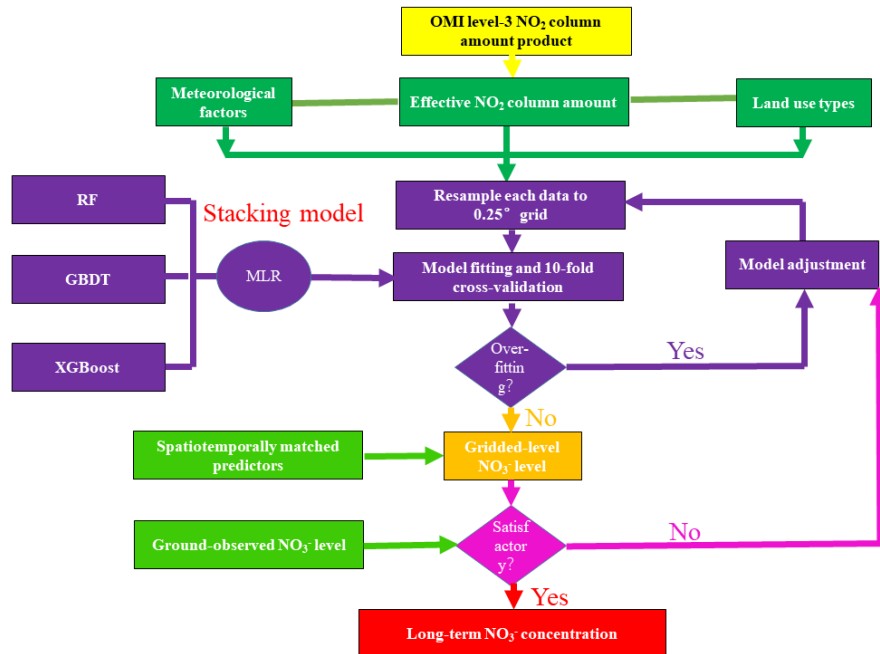


**Fig. 3** Density scatterplots of 10-fold cross-validation results for monthly NO$_3^-$ estimation (Unit:
μg/m$^3$) across China for the ensemble decision trees model including (a), RF (b), GBDT (c), and
XGBoost (d), respectively.

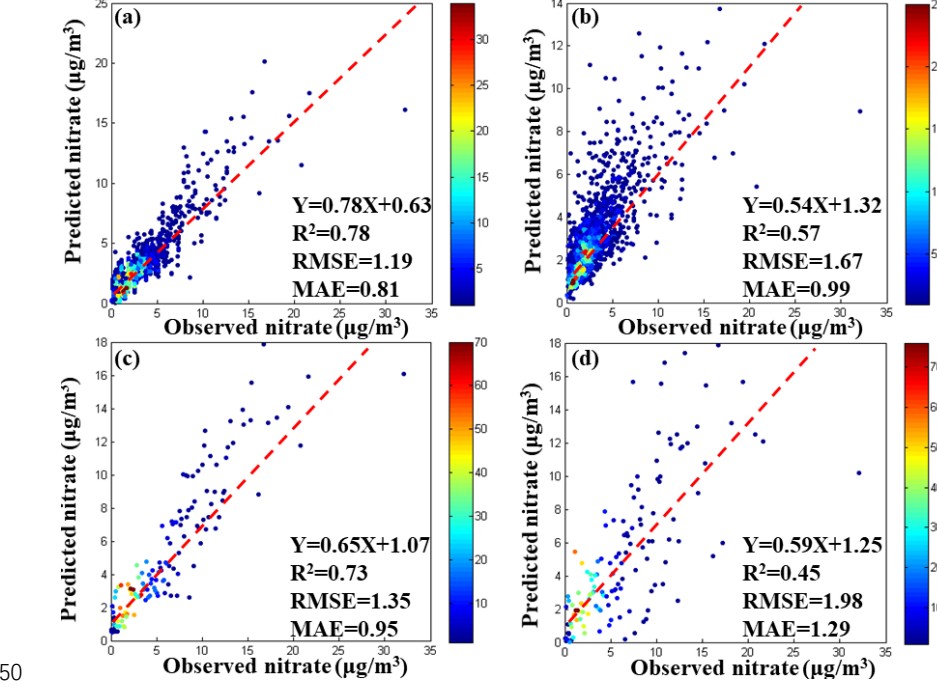


**Fig. 4** The transferability validation of the ensemble model in estimating NO$_3^-$ concentration over
China based on the unlearning observation data (Shen et al., 2013; Shen et al., 2009; Wang et al.,
2019a; Xu et al., 2018b). The linear regression curve is added in the figure. The blue square
represents the data points, and the red dashed line denotes the best-fit line through the data points.

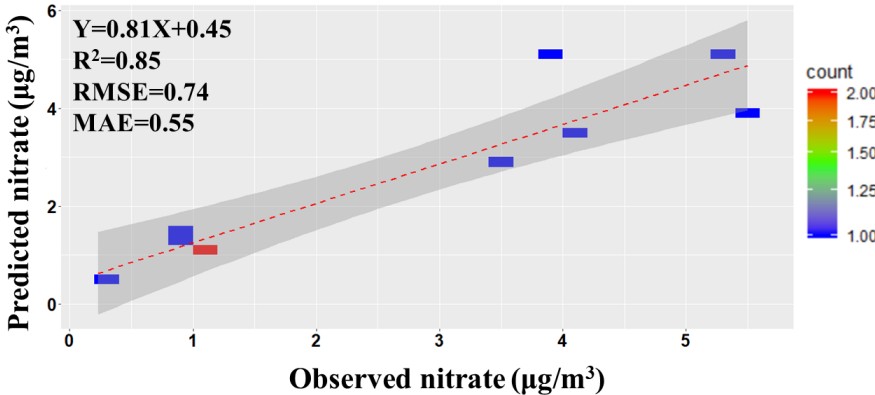




**Fig. 5** The spatial pattern of ambient $NO_3^-$ concentration ($\mu g/m^3$) over China during 2005-2015.

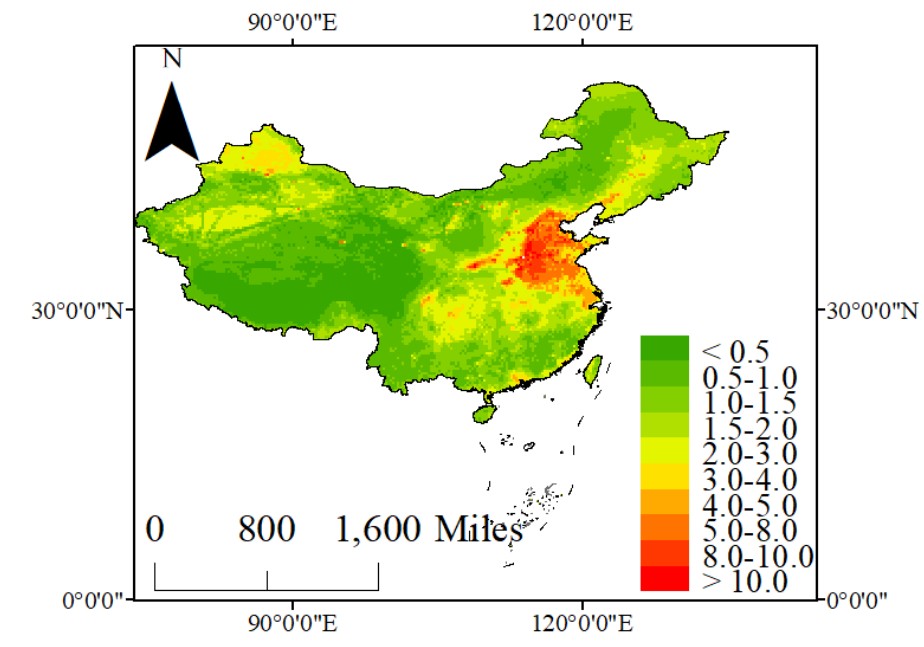






**Fig. 6** Satellite-derived annual mean NO₃⁻ concentration (μg/m³) across the entire China from (a)-(k) 2005-2015.

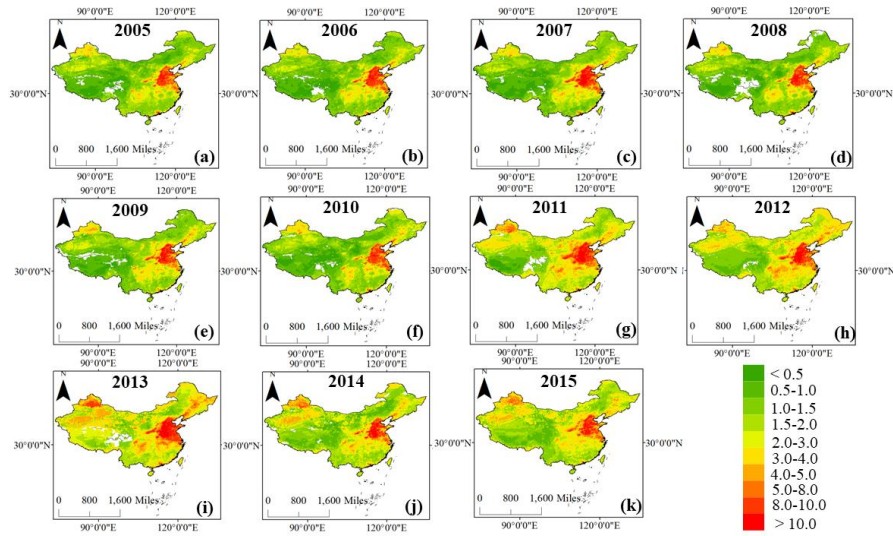





**Fig. 7** The annual mean NO$_3^-$ concentrations in major regions across China during 2005-2015. The
solid lines denote the mean NO$_3^-$ concentrations and the shadow represents the range of NO$_3^-$
concentrations.

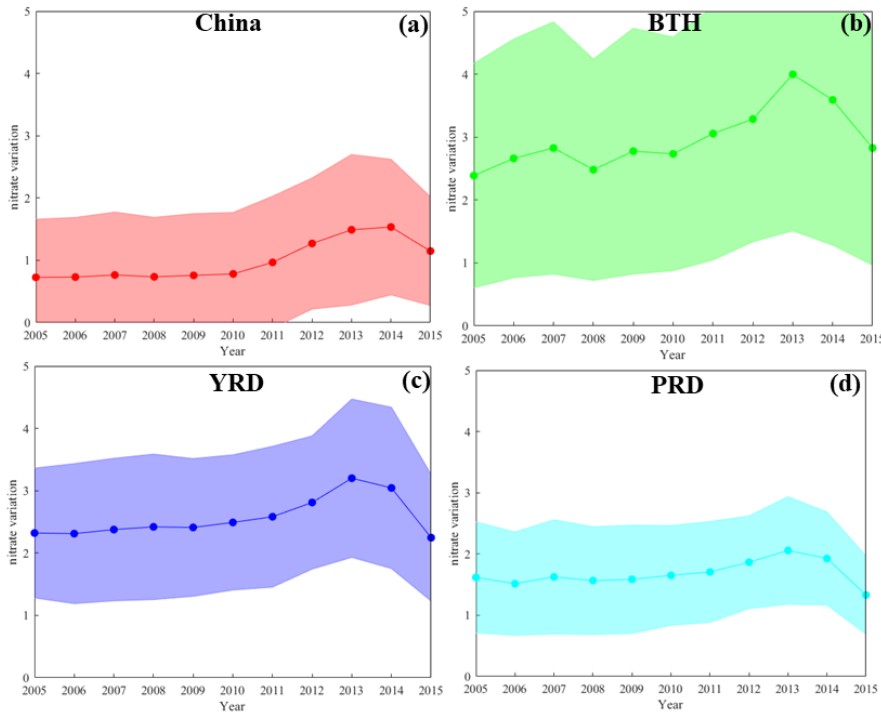



**Fig. 8** The long-term trends of $NO_3^-$ concentrations and significance levels in China (a, b, and c
denote the annual variation of ambient $NO_3^-$ concentration during 2005-2015, 2005-2014, and 2014-
2015, respectively. d, e, and f represent the significance level of $NO_3^-$ trend during these periods).
The pale green color denotes the regions with the significant variation of ambient $NO_3^-$
concentrations ($p < 0.05$), while the gray color represents the regions with insignificant variation of
$NO_3^-$ concentrations.

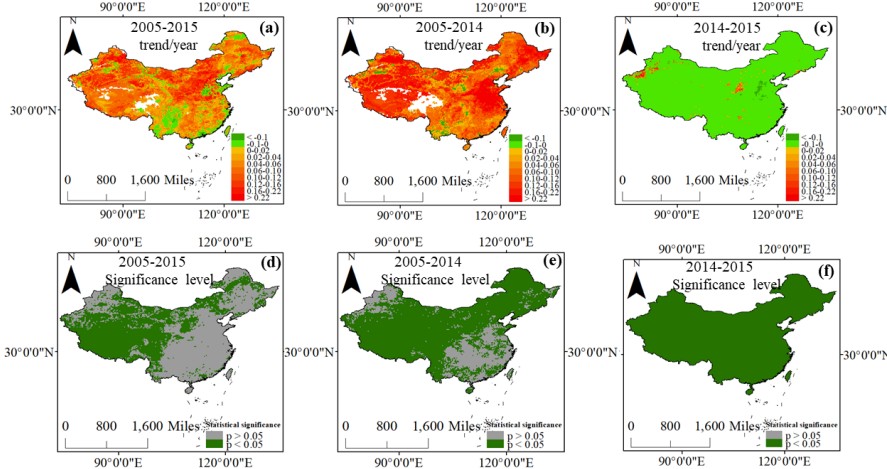




**Fig. 9** The mean concentrations of ambient $NO_3^-$ in spring (a), summer (b), autumn (c), and winter
(d) during 2005-2015 over China, respectively.

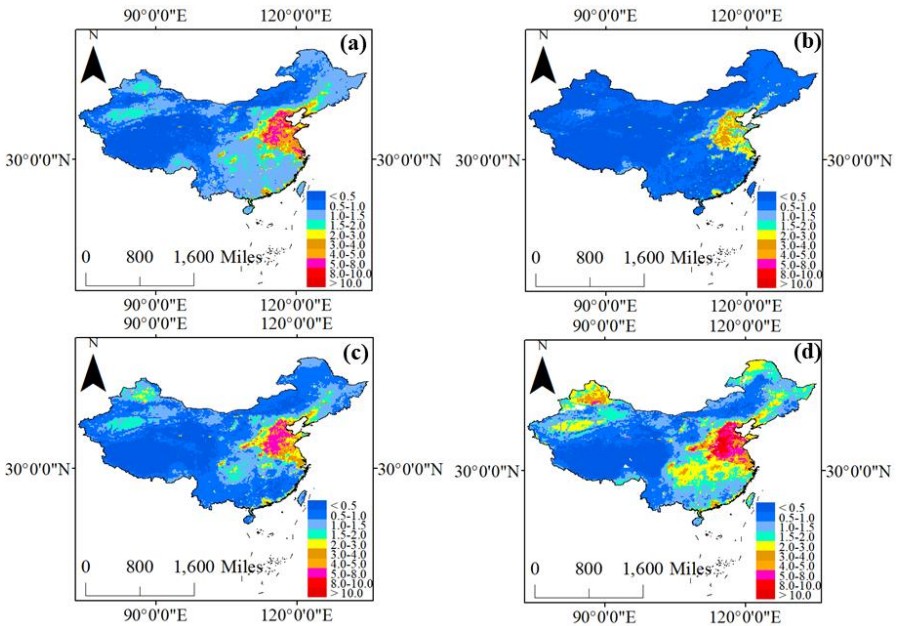






**Table 1** The cross-validation result of NO$_3^-$ estimation over China during 2010-2015.

| Season | R$^2$ value | Slope | RMSE (μg/m$^3$) | MAE (μg/m$^3$) |
|--------|-------------|-------|------------------|-----------------|
| 2010 | 0.62 | 0.60 | 1.39 | 0.90 |
| 2011 | 0.88 | 0.85 | 0.32 | 0.24 |
| 2012 | 0.89 | 0.86 | 0.33 | 0.28 |
| 2013 | 0.83 | 0.82 | 0.64 | 0.43 |
| 2014 | 0.74 | 0.76 | 1.50 | 1.04 |
| 2015 | 0.78 | 0.78 | 1.35 | 0.86 |






**Table 2** The cross-validation result of $NO_3^-$ estimation over China in four seasons.

| Season | $R^2$ value | Slope | RMSE ($\mu g/m^3$) | MAE ($\mu g/m^3$) |
|--------|-------------|-------|--------------------|--------------------|
| Spring | 0.80 | 0.80 | 0.71 | 0.48 |
| Summer | 0.85 | 0.84 | 0.29 | 0.20 |
| Autumn | 0.80 | 0.78 | 1.10 | 0.70 |
| Winter | 0.75 | 0.73 | 1.85 | 1.23 |






**Table 3** The cross-validation result of $NO_3^-$ estimation over China in different regions (Northeast
China includes Heilongjiang, Jilin, and Liaoning provinces; NCP includes Beijing, Tianjin, Hebei,
Henan, Shandong, and Shanxi provinces; Southeast China includes Jiangsu, Zhejiang, Fujian,
Guangdong, Jiangxi, Anhui, Hunan, Hainan, Shanghai, and Hubei provinces; Southwest China
includes Yunnan, Guangxi, Sichuan, Tibet, Chongqing, and Guizhou provinces; Northwest China
includes Inner Mongolia, Xinjiang, Gansu, Qinghai, Ningxia, and Shaanxi.

| Season | $R^2$ value | Slope | RMSE ($\mu g/m^3$) | MAE ($\mu g/m^3$) |
|---|---|---|---|---|
| Northeast China | 0.44 | 0.43 | 1.30 | 0.81 |
| NCP | 0.70 | 0.64 | 1.74 | 1.06 |
| Southeast China | 0.59 | 0.57 | 1.50 | 0.84 |
| Southwest China | 0.60 | 0.59 | 2.08 | 1.41 |
| Northwest China | 0.58 | 0.52 | 2.06 | 1.38 |




**Table 4** The trend analysis of $NO_3^-$ concentrations in China, BTH, YRD, and PRD regions during
788 2005-2015.

| Period | Trend | China | BTH | YRD | PRD |
|---|---|---|---|---|---|
| 2005-2014 | Trend (μg/m³/year) | 0.08 | 0.13 | 0.08 | 0.03 |
| | Significance | $p < 0.05$ | $p < 0.05$ | $p < 0.05$ | $p < 0.05$ |
| 2014-2015 | Trend (μg/m³/year) | -0.40 | -0.76 | -0.79 | -0.59 |
| | Significance | $p < 0.05$ | $p < 0.05$ | $p < 0.05$ | $p < 0.05$ |
| 2005-2015 | Trend (μg/m³/year) | 0.04 | 0.04 | -0.01 | -0.03 |
| | Significance | $p < 0.05$ | $p > 0.05$ | $p > 0.05$ | $p < 0.05$ |
