# Peer review of "Long-term trends of ambient nitrate ( $\text{NO}_3^-$ ) concentrations across China based on ensemble machine-learning models"

_Earth System Science Data, 2020_

## Short Comment (SC1) · 18 Jan 2021

In this study, the authors provided an ensemble model by stacking RF, GBDT, and XG-Boost to acquire monthly ambient nitrate concentrations over China. Generally, the topic of this study is very interesting since national-scale products of ambient chemical components are of great importance. However, the adoption of datasets in this paper is not convincing. To be specific, the spatial distribution of ground sites (only 32) is very sparse, which means that they do not cover most of the study area. How could the authors ensure the accuracy of the whole study area using these ground truths? I wonder how to validate the result in the regions without ground measure-

ments, such as Tibet. Such regions are numerous in this study. Besides, GEOS FP (http://wiki.seas.harvard.edu/geos-chem/index.php/GEOS-FP) can provide global 3-hour ambient nitrate concentrations at a similar spatial resolution. What is the main contribution of this study compared to GEOS FP? The authors need to justify the above issues in detail. Some minor comments are listed below. 1. Section 3: Why did the authors select these three machine learning methods for stacking? What if the authors only chose two of them? 2. Fig. 2: I notice that this flowchart is very similar to those in the authors' previous publications (e.g., Developing a novel hybrid model for the estimation of surface 8h ozone (O3) across the remote Tibetan Plateau during 2005–2018). Maybe a new style would be better. 3. Line 206: The parameters for RF, GBDT, and XGBoost are not given. Please provide them. 4. Fig. 3: XGBoost shows the worst performance, which is unusual. The authors need to provide some discussions. Did this happen in other literatures? 5. Fig. 5: Some point-shaped high values exist in the results (e.g, Northern China), which look like noises. Is this spatial distribution correct?

---

## Referee Comment (RC1) · Anonymous Referee #1 · 3 Feb 2021

General comment:

Based on surface observation, satellite product, meteorological data, land use types and other covariates, this research has developed a monthly NO3- dataset at 0.25° resolution over China during 2005-2015, using ensemble machine-learning models. The long-term NO3- dataset is valuable for the air pollution control work in China. Compared with previous products, this new method also shows better performance in predicting accuracy and inspires peers to utilize interdisciplinary approaches to solve environmental issues. However, I suggest some modifications are necessary before being accepted. My comments are as below:

1. More attention is needed to the details in scientific writing. For example, the abbreviation should have an explanation when it appeared at the first time, NNDMN (Line 127), ERA-Interim (Line 150)ïijŇAOD (Line 283) etc. Please check the manuscript carefully. Besides, U/V wind components are accurately latitudinal and meridional wind components in Line 153.

Figure 3, the name of the color bar is missing.

2. L134, The detection limit of particulate NO3 concentration over China is said to be 0.05 $\mu$g/m3 which is unlikely to be true. The authors needs to check for it.

3. Line 248-253 and Figure S3, what is the purpose to discuss the relationship between observed NO3- concentration and other parameters using Pearson correlation analysis?

3. In section 2.3, more details are needed about the method to assimilate the socioeconomic data (GDP, population etc.) every five years to the seasonal or monthly resolution.

3. Line 165, the importance values have been applied to select the independent variables to do the NO3- prediction in this research. The results of the importance values from the ensemble model and the selected variables are expected in this manuscript or in the supplement. In the same way, the regression coefficients (A, B, C, mentioned in Line 210) determined by the MLR model are expected, too. Because they are crucial parameters of the ensemble model.

4. To validate the excellent prediction performance of the ensemble model, detailed information about the observed data in Figure 4 are suggested to be labeled, such as sampling site, month, year etc. Data from sites far from the selected training sites and covering key areas will be more convincing and preferred.

5. A comparison also to chemical transport model results such as GEOS-FP (http://wiki.seas.harvard.edu/geos-chem/index.php/GEOS-FP) modeled nitrate will be

very valuable.

---

## Referee Comment (RC2) · Anonymous Referee #2 · 18 Feb 2021

In this study, Rui et al., attempt to use an ensemble of models to provide spatially and temporally resolved particulate nitrate concentrations in China. Of course, the availability of a reliable nitrate estimation across China would have immense value, but I see a number of issues in the presentation of the material. The sub-models are all poorly described and frankly quite confusing. The authors should clarify that they understand these beyond simply clicking the necessary buttons in a software package. Moreover, use of the explanatory variables is muddled. We don't know which variables were used and which thrown out due to redundancy, or how consistently that was done across sub-models or time periods. Finally the MLR statistics that define the ensemble model are not provided, including no discussion of multicollinearity issues. Application

of the ensemble model that followed is of course only relevant if the model itself is well established.

Based on Figure S1, the site in far northwest China only seems to have data from 2010-2011. This may dramatically impact the reliability of the method as that site is singularly located and likely unique in terms of the meteorological variable range, which the models are dependent on. Please comment on this data issue

Please provide information in the SI material and comment in the text about the monthly sample size of useable NO2 column data.

Sec 2.1: (i) These are described as monthly samples. Please clarify that they are continuously collected over a month and not a single 24-h sample each month. I believe the former is true, correct? (ii) you should include the reference to Xu et al., (2019) which reports NNDMN data from 2010-2015, (iii) within the Xu et al., (2019), they specify that the NO3 data is in units of ugN/mˆ3 and I wonder if you corrected for the molecular weights in this regard when you reported as NO3 concentrations? Your values seem similar to the ugN/mˆ3 values from Xu et al., (2019); (iv) finally, please specify the PM size-cutoff. This will impact whether nitrate from soil dust influenced the measurements

Xu, W., Zhang, L. & Liu, X. A database of atmospheric nitrogen concentration and deposition from the nationwide monitoring network in China. Sci Data 6, 51 (2019). https://doi.org/10.1038/s41597-019-0061-2

Sec. 2.3: with all of the variables introduced here, were they all incorporated into the MLR or just the sub-models? In the last paragraph of the section you mention, they are all incorporated into the model, but you don't indicate which model(s). Please clarify. And please provide detailed information about the finalized variables and which were found to be redundant. And was this made to be consistent for each time period? or the final predictor variables varied with time period analyzed?

Sec. 3.1 Equations 1-5: I (capital i), L, and chi are all not defined and generally make the use of this model too murky to understand

Equation 6: R is not defined; c would be better to have a subscript, perhaps naught?; what is "arg"?

Equation 8: again, undefined symbols and a general lack of description do not give the reader much confidence that the authors understand the complexities of the models that they are using

Sec. 4.2: I don't see the MLR statistics anywhere. Coefficients? Standard error of the coefficients? The MLR should also indicate whether or not it was really helpful to stack the models in the first place. You would expect this to suffer from multi-collinearity problems in that all three sub-models are attempting to predict the same nitrate concentrations. The authors can provide a variance inflation factor with interpretation in the text.

Figure 2: shouldn't each of the sub-models be connected with the original data tree (NO2 column / met data / land use) to show their optimization before incorporation into the MLR?

Figure 3: (i) something is missing from the caption. You seem to have forgotten to label what is 3a or 3d, (ii) please include sample size for each plot; (iii) why is there a different sample size for each plot?

Figure 4: (i) I don't understand why there is a color bar...is it for the number of points reported from another study? how can it be between 1 and 2?, (ii) the boxes are different sizes. Does it mean that these represent distributions of predicted NO3 concentrations? How to interpret the regression in this manner?

Figure 5: In the caption, Please specify that these values are from your ensemble model output

Figure 6: again, please specify these are from ensemble model output. "Satellitederived" could be mistaken for reanalysis data

Figure 7: (i) are these averaging all grid cells within a certain boundary? Are these boundaries specified anywhere? Please provide details in the text (ii) add the average measured trends for plots b,c, and d

Figure 8: plots c and f are confusing. For most of the map, you show a change of only -0.1 to 0 ug/m3 and with a significant trend!. Are these comparing two annual values each with n=12 at each grid cell? or 24 points in sequence? Even if the latter, it seems incredible that less than a 0.1 ug/m3 change tested significant over 24 points

Table 1: (i) should the first column be labeled "year"? or you intended to add more data to the table? (ii) somewhere you should indicate the sample size for the various years

Figure S1: plots should be labeled with year of data or with a letter and specified in the caption

Figure S3: all abbreviations should be described in the figure caption

Other comments:

Lines 76-79: this sentence is difficult to understand. Please revise

Lines 106-107: please expand this sentence to clarify your meaning or perhaps delete it

Lines 117-119: this sentence is an example where you should specify "particulate nitrate concentration" and there are numerous other places that you should indicate this to distinguish from a possible misinterpretation of rainwater or cloudwater nitrate, the former which is also measured at NNDMN sites

Line 127: NNDMN is not defined

Line 135: this reference should be specified 2018a or 2018b or both

Lines 151-154: this is not a complete sentence, please clarify your meaning

Line 206: "to determine the appropriate parameter" ..... does it mean a regression coefficient? or this is the 'c' value from the various models?

Lines 231-233: what does the first part of this sentence mean? "we only estimated the missing ratio of NO2 column..." please clarify

Lines 242-243: specify these as measured monthly NO3 concentrations

Line 249: I don't see urban land area designations in Fig S2

Lines 252-253: is there anything else to comment about this? any interpretation?

Lines 259-260: I don't understand why you say 'opposite trends' ....doesn't seem opposite. XGBoost is worst and ensemble best in all statistics

Line 269: "Furthermore," should be "However,"

Lines 277-279: I don't understand the latter half of this sentence. Please revise the text

Lines 280-281: is this for all years data combined? please specify

Lines 292-294: this sentence should start with "Although"

Lines 303-304: I don't understand this sentence. Please revise the text

Lines 304-306 and 307-308: Does this explain why RMSE and MAE are poor in these areas? I don't see the definite connection

Lines 306-307: are these areas in Northwest China? please clarify

Lines 330-331: "especially for hindcast of air pollutants" .... you mean compared to the forecast of air pollutants? Seems strange that a forecast model would have a better statistical result

Lines 365-368: the use of "speed" in these sentences is inappropriate. Example sentence revision for one of them: "For instance, the ambient NO3 level in BTH increased

remarkably by 0.13 $\mu$g/m3/year from 2005-2014."

Lines 391-400: decrease of SO2 emissions but not NOx emissions can further lead to NO3 increases because of decreased aerosol acidity, which is dictated by SO4 in particulate matter

Line 461: does this mean average available NO2 columns data was only ~43% each month? Or this was a minimum value and only at one site?

---

## Author Comment (AC1) · 27 Feb 2021

**Dear editor,**

Here we submitted our updated manuscript for consideration to be published on Earth System

**Science Data**

The further information about our manuscript is as follows:

**Topic:** Long-term trends of ambient nitrate (NO3-) concentrations across China based on ensemble machine-learning models

**Type of Manuscript: article**

Authors: Rui Lia, Lulu Cuia\*, Yilong Zhaoa, Wenhui Zhoua, Hongbo Fua,b,c\*

**Corresponding author:**

Hongbo Fu; Address: Department of Environmental Science and Engineering, Fudan University, Shanghai 200433, China; Tel.: (+86)21-5566-5189; Fax: (+86)21-6564-2080; Email: fuhb@fudan.edu.cn

Lulu Cui; 15110740004@fudan.edu.cn

**#Reviewer 1:** Based on surface observation, satellite product, meteorological data, land use types and other covariates, this research has developed a monthly NO3- dataset at 0.25° resolution over China during 2005-2015, using ensemble machine-learning models. The long-term NO3- dataset is valuable for the air pollution control work in China. Compared with previous products, this new method also shows better performance in predicting accuracy and inspires peers to utilize interdisciplinary approaches to solve environmental issues. However, I suggest some modifications are necessary before being accepted. My comments are as below:

**Comment 1:** More attention is needed to the details in scientific writing. For example, the abbreviation should have an explanation when it appeared at the first time, NNDMN (Line 127), ERA-Interim (Line 150), AOD (Line 283) etc. Please check the manuscript carefully. Besides, U/V wind components are accurately latitudinal and meridional wind components in Line 153. Figure 3, the name of the color bar is missing.

**Response:** Thank for reviewer's suggestions. The full names for all of the abbreviations appearing at the first time have been added in the revised version. Besides, the latitudinal and meridional wind components have been added instead of U/V wind components (Line 157-158). The name of the color bar denotes the sampling size for each model.

**Comment 2:** L134, The detection limit of particulate NO3- concentration over China is said to be

 $0.05 \,\mu g/m^3$  which is unlikely to be true. The authors needs to check for it.

**Response:** Thank for reviewer's suggestions. Based on the study in Xu et al. (2015 ACP) and Xu et al. (2019 Scientific data), the detection limit was determined as 0.01 mg N/L for NO3- (Line 164). The error has been corrected in the revised version.

**Comment 3:** Line 248-253 and Figure S3, what is the purpose to discuss the relationship between observed  $NO_3^-$  concentration and other parameters using Pearson correlation analysis? In section 2.3, more details are needed about the method to assimilate the socioeconomic data (GDP, population etc.) every five years to the seasonal or monthly resolution. Line 165, the importance values have been applied to select the independent variables to do the  $NO_3^-$  prediction in this research. The results of the importance values from the ensemble model and the selected variables are expected in this manuscript or in the supplement. In the same way, the regression coefficients (A, B, C, mentioned in Line 210) determined by the MLR model are expected, too. Because they are crucial parameters of the ensemble model.

**Response:** Thank for reviewer's suggestions. The correlation analysis of observed  $NO_3^-$  concentration and independent variables were performed to examine the possible factors for the spatiotemporal variability of  $NO_3^-$  concentrations. It is only a tentative procedure to develop the machine-learning model. Due to coarse temporal resolution of socioeconomic data (e.g., GDP), we have to use some methods to resample these data to the month resolution. First of all, population density (PD) and GDP in 1995, 2000, 2005, 2010, and 2015 were linearly interpolated to calculate PD and GDP in each year during 2005-2015. Then, the yearly GDP data were uniformly divided by 12 to estimate the monthly GDP. The monthly PD equaled to the corresponding yearly PD. In our study, socioeconomic data were allocated to the month scale in a very simple way because the monthly GDP in each city were not available. The importance values of the ensemble model have been shown in Fig. S7. In the final model, all of the variables except GDP, PD, and grassland have been applied to estimate the ambient  $NO_3^-$  concentrations across China. The regression coefficients including A, B, C, and the residual error ( $e_{ij}$ ) determined by the MLR model were 0.42, 0.77, 0.09, and -0.87, respectively.

**Comment 4:** To validate the excellent prediction performance of the ensemble model, detailed information about the observed data in Figure 4 are suggested to be labeled, such as sampling site, month, year etc. Data from sites far from the selected training sites and covering key areas will be

more convincing and preferred.

**Response:** I agree with reviewer's suggestions. In our study, we used some unlearned data collected from previous references to validate the performance of the developed model. The detailed site information including latitude, longitude, year, and month have been shown in Table S2. Although most of these sites were located in East China, these sites were not close to the training sites and even several sites were located in Tibetan Plateau. Meanwhile, the observation data during 2006-2008 were also used to validate the temporal transferability of this model. Moreover, the site-based cross-validation was also applied to test the transferability of this model (Line 339-346). The overall result suggested the model (dataset) showed robust performance.

**Comment 5:** A comparison also to chemical transport model results such as GEOS-FP (http://wiki.seas.harvard.edu/geos-chem/index.php/GEOS-FP) modeled nitrate will be very valuable.

**Response:** Thank for reviewer's suggestions. We also want to compare the modelled nitrate in our study with the GEOS-FP nitrate product. Unfortunately, we only found global nitrate concentrations since December, 2017, which cannot be compared with the nitrate concentrations in 2005-2015 retrieved by our study. In the future work, we hope to estimate the recent nitrate concentration after 2017 and then compare with the GEOS-FP product when the updated observation data are available.

**#Reviewer 2:**

In this study, Rui et al., attempt to use an ensemble of models to provide spatially and temporally resolved particulate nitrate concentrations in China. Of course, the availability of a reliable nitrate estimation across China would have immense value, but I see a number of issues in the presentation of the material. The sub-models are all poorly described and frankly quite confusing. The authors should clarify that they understand these beyond simply clicking the necessary buttons in a software package. Moreover, use of the explanatory variables is muddled. We don't know which variables were used and which thrown out due to redundancy, or how consistently that was done across sub-models or time periods. Finally the MLR statistics that define the ensemble model are not provided, including no discussion of multicollinearity issues. Application of the ensemble model that followed is of course only relevant if the model itself is well established.

**Response:** Thank for reviewer's suggestions. We have significantly revised the manuscript based on reviewer's suggestions. The detailed description of the ensemble model has been added in the

revised version. Besides, some new methods have been added to validate the robustness of the ensemble model and the NO3- dataset.

**Comment 1:** Based on Figure S1, the site in far northwest China only seems to have data from 2010-2011. This may dramatically impact the reliability of the method as that site is singularly located and likely unique in terms of the meteorological variable range, which the models are dependent on. Please comment on this data issue.

**Response:** Thank for reviewer's suggestions. Indeed, the site in Northwest China only has data during 2010-2011 because the long-term monitoring in the arid region is a hard task. To date, no long-term  $NO_3^-$  dataset in Northwest China during 2012-2015. However, the lack of  $NO_3^-$  dataset during this period did not significantly degrade the modelling performance of  $NO_3^-$  estimates. In the revised version, we performed the site-based cross validation to demonstrate the spatial transferability of this ensemble model. The basic principle is that all of the sites were evenly classified into ten clusters based on the geographical locations. Afterwards, nine of ten were used to train the model and then test the model based on the remained one. After ten round, all of the observed values versus estimate values was considered to be the final result to validate the spatial transferability of this model. Based on the site-based cross validation, we found that the ensemble model showed the better transferability in predicting the  $NO_3^-$  concentrations. Thus, the estimated  $NO_3^-$  concentration in Northwest China could be considered to be reliable though the measured  $NO_3^-$  dataset in this region since 2012 was missing.

**Comment 2:** Please provide information in the SI material and comment in the text about the monthly sample size of useable NO2 column data.

**Response:** Thank for reviewer's suggestions. A total of  $1636 \text{ NO}_2$  column data during 2011-2015 was applied to develop the ensemble model to estimate the national NO3- concentration. Afterwards, 1554236 useable NO2 column data during 2005-2015 was applied to predict the gridded monthly NO3- concentrations across China. The detailed information has been added in the revised version (Table S1).

**Comment 3:** Sec 2.1: (i) These are described as monthly samples. Please clarify that they are continuously collected over a month and not a single 24-h sample each month. I believe the former is true, correct? (ii) you should include the reference to Xu et al., (2019) which reports NNDMN data from 2010-2015, (iii) within the Xu et al., (2019), they specify that the NO3 data is in units of

ugN/m3 and I wonder if you corrected for the molecular weights in this regard when you reported as NO3 concentrations? Your values seem similar to the ugN/m3 values from Xu et al., (2019); (iv) finally, please specify the PM size-cutoff. This will impact whether nitrate from soil dust influenced the measurements. Xu, W., Zhang, L. & Liu, X. A database of atmospheric nitrogen concentration and deposition from the nationwide monitoring network in China. Sci Data 6, 51 (2019). https://doi.org/10.1038/s41597-019-0061-2

**Response:** Thank for reviewer's suggestions. We have added the description about the samples were continuously collected over a month in the revised version (Line 138). Besides, the reference of Xu et al. (2019) has been cited in the paper. The unit of  $\mu$ g/m3 has been changed into  $\mu$ g N m-3. The PM size-cutoff of NO3- particles collected from the nationwide monitoring network was PM10 (Line 136).

**Comment 4:** Sec. 2.3: with all of the variables introduced here, were they all incorporated into the MLR or just the sub-models? In the last paragraph of the section you mention, they are all incorporated into the model, but you don't indicate which model(s). Please clarify. And please provide detailed information about the finalized variables and which were found to be redundant. And was this made to be consistent for each time period? Or the final predictor variables varied with time period analyzed?

**Response:** Thank for reviewer's suggestions. The independent variables were incorporated into the sub-models rather than MLR. The simulated  $NO_3^-$  concentrations for three sub-models were then input into the MLR to further estimate the  $NO_3^-$  concentrations based on the ground-level  $NO_3^-$  concentrations. The method is a two-stage model, which could enhance the modelling performance of  $NO_3^-$  estimates. At first, all of the variables were input into the machine-learning model and then to remove some redundant predictors.

All of the independent variables were input into the sub-model rather than MLR. The simulated NO3- levels from RF, GBDT, and XGBoost were integrated into the MLR model, which was the second-stage procedure.

Finally, all of the variables except GDP, PD, and grassland have been applied to estimate the ambient NO3- concentrations across China.

In our study, we did not consider the time variability of the importance of independent variables for the  $NO_3^-$  estimates because the training sample size was not large. However, the time (month of

year) was incorporated into the training model as a key variable because the time autocorrelation of  $NO_3^-$  levels in the adjacent month might be paid more attention. As shown in Fig. S7, the time autocorrelation was not the most important variable compared with  $NO_2$  column, and thus we did not need to analyze the time variability of the importance of independent variables for the  $NO_3^-$  estimates.

**Comment 5:** Sec. 3.1 Equations 1-5: I (capital i), L, and chi are all not defined and generally make the use of this model too murky to understand

**Response:** Thank for reviewer's suggestions. where  $(x_i, y_i)$  denotes the sample for i = 1, 2, ..., N in M regions  $(M_1, M_2, ..., M_z)$ ; *I* denotes the weight of each branch; L denotes the branch of decision tree;  $c_m$  represents the response to the model;  $c_z^{A}$  denotes the best value, m represents the feature variable;  $c_1$  denotes the mean value of left branch;  $c_2$  denotes the mean value of right branch; n is the split point. The detailed revision is shown in the body text.

**Comment 6:** Equation 6: R is not defined; c would be better to have a subscript, perhaps naught?; what is "arg"?

**Response:** Thank for reviewer's suggestions.  $R_{ij}$  denotes each leaf node for the decision trees. c is regarded as the optimal value when  $c_{ij}$  reaches the least value, which is not a naught. "argmin" denotes that the least value of the equation of  $\sum_{xi \in R_{ij}} L(y_i, f_{i-1}(x_i) + c)$ , which equals to minimum.

**Comment 7:** Equation 8: again, undefined symbols and a general lack of description do not give the reader much confidence that the authors understand the complexities of the models that they are using

**Response:** Thank for reviewer's suggestions. It is a negligence in our study. All of the symbols were defined in the equation 8 in the revised version.

**Comment 8:** Sec. 4.2: I don't see the MLR statistics anywhere. Coefficients? Standard error of the coefficients? The MLR should also indicate whether or not it was really helpful to stack the models in the first place. You would expect this to suffer from multi-collinearity problems in that all three sub-models are attempting to predict the same nitrate concentrations. The authors can provide a variance inflation factor with interpretation in the text.

**Response:** Thank for reviewer's suggestions. The regression coefficients including A, B, C, and the residual error (eij) determined by the MLR model were 0.42, 0.77, 0.09, and -0.87, respectively. The

standard error of A, B, and C were 0.02, 0.03, and 0.01, respectively. We employed the stacking model of three machine-learning models because they could achieve the better performance compared with the individual model or the two-stage models. As shown in the following figure, we found that the  $R^2$  values of RF+GBDT, GBDT+XGBoost, and RF+XGBoost models were lower than the stacking model of three machine-learning algorithms (see the following figure). Furthermore, both of RMSE and MAE for two-stage models were higher than those of stacking model of three machine-learning algorithms. Based on the result, the stacking model of RF, GBDT, and XGBoost were applied to estimate the ambient  $NO_3^-$  concentrations across China. Furthermore, the MLR model did not suffered from significant multi-collinearity problems. The variance inflation factors of RF (2.01), GBDT (2.69), and XGBoost (2.08) were significantly lower than 10, which suggested the MLR model was robust.

**Comment 9:** Figure 2: shouldn't each of the sub-models be connected with the original data tree (NO2 column / met data / land use) to show their optimization before incorporation into the MLR? **Response:** Thank for reviewer's suggestions. All of the hyperparameters were adapted to ensure the optimization of each sub-model, and then the simulated  $NO_3^-$  concentrations by each sub-model was integrated into the MLR model to further estimate the  $NO_3^-$  concentration. The ensemble model is

two-stage model, and MLR model is an optimized process. All of the independent variables including NO2 column, meteorological conditions were only incorporated into the sub-model instead of MLR model. In the MLR model, only the simulated values by RF, GBDT, and XGBoost were input.

**Comment 10:** Figure 3: (i) something is missing from the caption. You seem to have forgotten to label what is 3a or 3d, (ii) please include sample size for each plot; (iii) why is there a different sample size for each plot?

**Response:** Thank for reviewer's suggestions. We have redrawn the figure 3 in the revised version and the caption has been significantly revised. The sample size of each plot is 1636. Besides, the sample size of each plot has been shown in the same colorbar.

**Comment 11:** Figure 4: (i) I don't understand why there is a color bar...is it for the number of points reported from another study? how can it be between 1 and 2?, (ii) the boxes are different sizes. Does it mean that these represent distributions of predicted NO3 concentrations? How to interpret the regression in this manner?

**Response:** Thank for reviewer's suggestions. Yes, the colorbar denotes the number of points. Figure 4 reflects the correlation of observed  $NO_3^-$  concentrations collected from previous studies and corresponding simulated ones in the same grid and during the same period. These data were collected from previous studies to validate the transferability of the ensemble model because these data obtained from previous studies were not used to develop the model, which can be treated as the unlearned dataset. We have redrawn Figure 4 and ensured the same size of these points. The regression curve in this figure represents the optimal fitting curve of estimated  $NO_3^-$  levels and observed ones.

**Comment 12:** Figure 5: In the caption, Please specify that these values are from your ensemble model output

**Response:** I agree with reviewer's suggestions. I have rewritten the caption.

**Comment 13:** Figure 6: again, please specify these are from ensemble model output. "Satellitederived" could be mistaken for reanalysis data

**Response:** I agree with reviewer's suggestions. I have rewritten the caption.

**Comment 14:** Figure 7: (i) are these averaging all grid cells within a certain boundary? Are these boundaries specified anywhere? Please provide details in the text (ii) add the average measured

trends for plots b,c, and d

**Response:** Thank for reviewer's suggestions. Indeed, the inter-annual NO3- concentrations in China, BTH, YRD, and PRD were estimated based on the average values of NO3- concentrations within a certain boundary. The detailed information has been added in Fig. S1. The ambient NO3- level in BTH showed the remarkable increase during 2005-2013 by 0.20  $\mu$ g/m3/year. Afterwards, the NO3- level decreased rapidly from 2013 to 2015 at a rate of -0.58  $\mu$ g/m3/year. The NO3- concentrations in YRD (0.11  $\mu$ g/m3/year) and PRD (0.05  $\mu$ g/m3/year) both showed the slight increases during 2005-2013, though the statistical test revealed the increases were significant. However, the NO3- concentrations in YRD and PRD showed the dramatic decreases with -0.48 and -0.36  $\mu$ g/m3/year during 2013-2015, respectively.

**Comment 15:** Figure 8: plots c and f are confusing. For most of the map, you show a change of only -0.1 to 0 ug/m3 and with a significant trend!. Are these comparing two annual values each with n=12 at each grid cell? or 24 points in sequence? Even if the latter, it seems incredible that less than a 0.1 ug/m3 change tested significant over 24 points

**Response:** Thank for reviewer's suggestions. It is a big fault in our study. The colorbar in Fig. 8c is misinformed. We have corrected the errors in the revised version. For the significance test of  $NO_3^-$  trends, we used Mann-Kendall method to perform the trend analysis of  $NO_3^-$  concentration, which could test whether the  $NO_3^-$  concentration suffered from the significant trend during some periods. The method is a very old statistical technique developed by Mann (1945) and Kendall (1975) and has been applied in many fields (e.g., hydrology, atmospheric environment, meteorology). We performed the analysis based on the simple equation summarized in these references.

**Comment 16:** Table 1: (i) should the first column be labeled "year"? or you intended to add more data to the table? (ii) somewhere you should indicate the sample size for the various years

**Response:** Thank for reviewer's suggestions. The "season" has been changed into "year". The sampling size has been added in Table 1-3.

**Comment 17:** Figure S1: plots should be labeled with year of data or with a letter and specified in the caption

**Response:** I agree with reviewer's suggestions. The year of data has been added in the plots (Fig. S2).

Comment 18: Figure S3: all abbreviations should be described in the figure caption

**Response:** I agree with reviewer's suggestions. The full name of all the variables were added in the revised version (Fig. S4).

Other comments:

Comment 19: Lines 76-79: this sentence is difficult to understand. Please revise

**Response:** I agree with reviewer's suggestions. The sentence has been changed into "these sparse ground-observed sites did not accurately reflect the high-resolution  $NO_3^-$  pollution especially the regions far away from these sites because each station only possessed limited spatial representative and  $NO_3^-$  concentration was often highly variable in space and time" (Line 78-80).

**Comment 20:** Lines 106-107: please expand this sentence to clarify your meaning or perhaps delete it

**Response:** I agree with reviewer's suggestions. The sentence has been deleted.

**Comment 21:** Lines 117-119: this sentence is an example where you should specify "particulate nitrate concentration" and there are numerous other places that you should indicate this to distinguish from a possible misinterpretation of rainwater or cloudwater nitrate, the former which is also measured at NNDMN sites

**Response:** Thank for reviewer's suggestions. The particulate nitrate concentration has been emphasized in some other places to avoid the misinterpretation.

Comment 22: Line 127: NNDMN is not defined

**Response:** Thank for reviewer's suggestions. The full name of NNDMN is nationwide nitrogen deposition monitoring network (Line 129-130).

Comment 23: Line 135: this reference should be specified 2018a or 2018b or both

**Response:** I agree with reviewer's suggestions. Xu et al. (2018a) and Xu et al. (2019) were added in the revised version.

Comment 24: Lines 151-154: this is not a complete sentence, please clarify your meaning

**Response:** Thank for reviewer's suggestions. The sentence has been changed into "Among all of the daily meteorological data in ECMWF website, 2-m temperature  $(T_{2m})$ , 2-m dewpoint temperature  $(D_{2m})$ , 10-m latitudinal wind component  $(U_{10})$ , 10-m meridional wind component  $(V_{10})$ , sunshine duration (Sund), surface pressure (Sp), boundary layer height (BLH), and total precipitation (Tp) were applied to estimate national NO3- levels" (Line 157-160).

Comment 25: Line 206: "to determine the appropriate parameter" ..... does it mean a regression

coefficient? or this is the 'c' value from the various models?

**Response:** Thank for reviewer's suggestions. These parameters denote the hyperparameters of each decision tree model. The maximum depth, minimum leaf node size, number of parameters for split, and number of trees of RF model was 8, 25, 6, and 500, respectively. The maximum depth, min\_samples\_split, and subsample ratio reached 7, 300, 0.7, respectively. The maximum depth, minimum child weight, and subsample ratio of XGBoost was 9, 5, and 0.7, respectively.

**Comment 26:** Lines 231-233: what does the first part of this sentence mean? "we only estimated the missing ratio of  $NO_2$  column..." please clarify

**Response:** Thank for reviewer's suggestions. Satellite-based  $NO_2$  columns generally suffered from some missing values, which were difficult to fill because the dataset of ground-observed  $NO_2$  columns was not open access. These missing values might increase the uncertainty of  $NO_3^-$  estimates. **Comment 27:** Lines 242-243: specify these as measured monthly  $NO_3^-$  concentrations

**Response:** I agree with reviewer's suggestions. The "ground-observed  $NO_3^-$  concentrations" has been added in the revised version (Line 257).

Comment 28: Line 249: I don't see urban land area designations in Fig S2

**Response:** Thank for reviewer's suggestions. The "urban land area" has been deleted.

**Comment 29:** Lines 252-253: is there anything else to comment about this? any interpretation? **Response:** Thank for reviewer's suggestions. The correlation analysis here is only to analyze the possible relationship between independent variables and  $NO_3^-$  level, which is an exploratory analysis.

**Comment 30:** Lines 259-260: I don't understand why you say 'opposite trends' ....doesn't seem opposite. XGBoost is worst and ensemble best in all statistics

**Response:** Thank for reviewer's suggestions. The  $R^2$  value showed the order of ensemble model > GBDT > RF > XGBoost, while the RMSE and MAE values followed the order of XGBoost > RF > GBDT > ensemble model. Thus, we believed that the  $R^2$  value showed the opposite order with RMSE and MAE.

Comment 31: Line 269: "Furthermore," should be "However,"

**Response:** I agree with reviewer's suggestions. "Furthermore" has been replaced by "However" (Line 291).

Comment 32: Lines 277-279: I don't understand the latter half of this sentence. Please revise the

**Response:** Thank for reviewer's suggestions. The sentence has been changed into "The higher RMSE and MAE observed in 2010 might be contributed by the relatively scarce training samples, while the higher RMSE and MAE likely attained to the higher  $NO_3$ - levels during other years."

In fact, both of the relatively poor performance and high  $NO_3^-$  levels caused the higher RMSE and MAE values. The sentence means the higher RMSE and MAE in 2010 was attributable to few training samples, while those in other years might be attributable to the higher  $NO_3^-$  levels (Line 300-302).

Comment 33: Lines 280-281: is this for all years data combined? please specify

**Response:** Thank for reviewer's suggestions. Indeed, the seasonal performance was conducted based on all years data.

Comment 34: Lines 292-294: this sentence should start with "Although"

Response: Thank for reviewer's suggestions. The sentence starts with "Although".

Comment 35: Lines 303-304: I don't understand this sentence. Please revise the text

**Response:** Thank for reviewer's suggestions. The sentence has been changed into "At first, the predictive performances of Southwest China and Northwest China were significantly worse than that of NCP, thereby leading to the higher RMSE and MAE" (Line 326-327).

**Comment 36:** Lines 304-306 and 307-308: Does this explain why RMSE and MAE are poor in these areas? I don't see the definite connection

**Response:** Thank for reviewer's suggestions. The sample size is a very important factor for the accuracy of  $NO_3^-$  estimates. Both of Northeast China and Northwest China possessed limited training samples (< 200), while the predictive performance of  $NO_3^-$  estimates in Northwest China was significantly better than those in Northeast China. It was assumed that the sampling sites in Northeast China were very centralized, while the sampling sites in Northwest China were uniformly distributed across the whole region. Some previous studies have confirmed that the spatial distribution of monitoring sites significantly affected the  $NO_3^-$  estimates for machine-learning models. In general, the uniform distribution of sites was beneficial to elevate the modelling accuracy of machine-learning models compared with the centralized distribution.

Comment 37: Lines 306-307: are these areas in Northwest China? please clarify

**Response:** Thank for reviewer's suggestions. Yes, these sites were located in Northwest China.

text

**Comment 38:** Lines 330-331: "especially for hindcast of air pollutants" .... you mean compared to the forecast of air pollutants? Seems strange that a forecast model would have a better statistical result

**Response:** Thank for reviewer's suggestions. We did not want to compare with the forecast result in this study. Thus, "hindcast" has been changed into "estimates" (Line 361).

**Comment 39:** Lines 365-368: the use of "speed" in these sentences is inappropriate. Example sentence revision for one of them: "For instance, the ambient  $NO_3^-$  level in BTH increased remarkably by 0.13  $\mu$ g/m3/year from 2005-2014."

**Response:** Thank for reviewer's suggestions. We have significantly revised the similar sentences in this paragraph based on the reviewer's suggestions (section 4.4).

**Comment 40:** Lines 391-400: decrease of SO2 emissions but not NOx emissions can further lead to  $NO_3^-$  increases because of decreased aerosol acidity, which is dictated by SO42- in particulate matter. **Response:** I agree with reviewer's suggestions. The explanation has been added in the revised version (Line 439-441).

**Comment 41:** Line 461: does this mean average available NO2 columns data was only  $\sim$ 43% each month? Or this was a minimum value and only at one site?

**Response:** Thank for reviewer's suggestions. The sentence means that number of available daily (not monthly) NO2 columns across China during 2005-2015 accounts for about 43% of all the daygrids ( $365 \times 11 \times 16320$ ), which included the NO2 columns in all of the grids ( $0.25^{\circ}$ ) rather than those in the sites.

**#SC Reviewer:** In this study, the authors provided an ensemble model by stacking RF, GBDT, and XGBoost to acquire monthly ambient nitrate concentrations over China. Generally, the topic of this study is very interesting since national-scale products of ambient chemical components are of great importance. However, the adoption of datasets in this paper is not convincing.

**Response:** Thank for reviewer's suggestions. We have significantly revised the manuscript based on reviewer's suggestions. Meanwhile, many validation method has been applied to confirm the robustness of the ensemble model and the reliability of our dataset.

**Comment 1:** To be specific, the spatial distribution of ground sites (only 32) is very sparse, which means that they do not cover most of the study area. How could the authors ensure the accuracy of the whole study area using these ground truths? I wonder how to validate the result in the regions

without ground measurements, such as Tibet. Such regions are numerous in this study.

**Response:** Thank for reviewer's suggestions. In our study, some methods have been applied to confirm the robustness of this model. First of all, some unlearned data collected from previous references have been used to test the transferability of this model (Fig. 4). The result suggested that the  $R^2$  value based on unlearned data was even higher than that of the developed model, indicating high accuracy of this dataset. This method might show some limitations due to scarce testing samples. Thus, we also employed a site-based cross validation method to examine the transferability of the ensemble model. The basic principle is that all of the sites were evenly classified into ten clusters based on the geographical locations. After ten round, all of the observed values versus estimate values was considered to be the final result to validate the spatial transferability of this model. As depicted in Fig. S6, the site-based cross-validation  $R^2$  value reached 0.73, which was slightly lower than the cross-validation  $R^2$  value of the training model (0.78).

Indeed, some regions such as Tibet lacks of monitoring sites due to the difficulty of long-term observation, we cannot confirm the accuracy of this dataset directly. However, the site-based cross-validation and the validation based on some unlearned data have suggested that the dataset is reliable. In the future work, we will train a new model to further enhance the accuracy of  $NO_3^-$  estimates when more samples are available.

**Comment 2:** Besides, GEOS-FP (http://wiki.seas.harvard.edu/geos-chem/index.php/GEOS-FP) can provide global 3-hour ambient nitrate concentrations at a similar spatial resolution. What is the main contribution of this study compared to GEOS FP? The authors need to justify the above issues in detail.

**Response:** Thank for reviewer's suggestions. We have tried our best to search the global 3-hour nitrate product in GEOS-FP website. Unfortunately, we only found the 3-hour nitrate concentrations since December, 2017. However, the long-term ambient nitrate concentrations during 2005-2015 were not available. The major contribution of our study is to obtain a long-term ambient nitrate concentrations across China and share with these data at an open website. Moreover, the data quality is convincing because the ground-level observation data has been assimilated into the model and the final model showed the robust performance. By contrast, the GEOS-FP product did not assimilate the ground-level NO3- concentrations in China, which might significantly biased from

the real situation in China due to the uncertainty of emission inventory and imperfect mechanisms of chemical transport model

**Comment 3:** Some minor comments are listed below. Section 3: Why did the authors select these three machine learning methods for stacking? What if the authors only chose two of them?

**Response:** Thank for reviewer's suggestions. We employed the stacking model of three machinelearning models because they could achieve the better performance compared with the individual model or the two-stage models. As shown in the following figure, we found that the R2 values of RF+GBDT, GBDT+XGBoost, and RF+XGBoost models were lower than the stacking model of three machine-learning algorithms. Furthermore, both of RMSE and MAE for two-stage models were higher than those of stacking model of three machine-learning algorithms. Based on the result, the stacking model of RF, GBDT, and XGBoost were applied to estimate the ambient NO3- concentrations across China.

---

## Author Comment (AC2) · 27 Feb 2021

**Dear editor,**

Here we submitted our updated manuscript for consideration to be published on **Earth System Science Data**

The further information about our manuscript is as follows:

**Topic:** Long-term trends of ambient nitrate ($NO_3^-$) concentrations across China based on ensemble machine-learning models

**Type of Manuscript:** article

**Authors:** Rui Li[a], Lulu Cui[a *], Yilong Zhao[a], Wenhui Zhou[a], Hongbo Fu[a,b,c *]

**Corresponding author:**

Hongbo Fu; Address: Department of Environmental Science and Engineering, Fudan University, Shanghai 200433, China; Tel.: (+86)21-5566-5189; Fax: (+86)21-6564-2080; Email: fuhb@fudan.edu.cn

Lulu Cui; 15110740004@fudan.edu.cn

**#Reviewer 1:** Based on surface observation, satellite product, meteorological data, land use types and other covariates, this research has developed a monthly $NO_3^-$ dataset at 0.25∘ resolution over China during 2005-2015, using ensemble machine-learning models. The long-term $NO_3^-$ dataset is valuable for the air pollution control work in China. Compared with previous products, this new method also shows better performance in predicting accuracy and inspires peers to utilize interdisciplinary approaches to solve environmental issues. However, I suggest some modifications are necessary before being accepted. My comments are as below:

**Comment 1:** More attention is needed to the details in scientific writing. For example, the abbreviation should have an explanation when it appeared at the first time, NNDMN (Line 127), ERA-Interim (Line 150), AOD (Line 283) etc. Please check the manuscript carefully. Besides, U/V wind components are accurately latitudinal and meridional wind components in Line 153. Figure 3, the name of the color bar is missing.

**Response:** Thank for reviewer's suggestions. The full names for all of the abbreviations appearing at the first time have been added in the revised version. Besides, the latitudinal and meridional wind components have been added instead of U/V wind components (Line 157-158). The name of the color bar denotes the sampling size for each model.

**Comment 2:** L134, The detection limit of particulate $NO_3^-$ concentration over China is said to be

0.05 μg/m$^3$ which is unlikely to be true. The authors needs to check for it.

**Response:** Thank for reviewer's suggestions. Based on the study in Xu et al. (2015 ACP) and Xu et al. (2019 Scientific data), the detection limit was determined as 0.01 mg N/L for NO$_3^-$ (Line 164). The error has been corrected in the revised version.

**Comment 3:** Line 248-253 and Figure S3, what is the purpose to discuss the relationship between observed NO$_3^-$ concentration and other parameters using Pearson correlation analysis? In section 2.3, more details are needed about the method to assimilate the socioeconomic data (GDP, population etc.) every five years to the seasonal or monthly resolution. Line 165, the importance values have been applied to select the independent variables to do the NO$_3^-$ prediction in this research. The results of the importance values from the ensemble model and the selected variables are expected in this manuscript or in the supplement. In the same way, the regression coefficients (A, B, C, mentioned in Line 210) determined by the MLR model are expected, too. Because they are crucial parameters of the ensemble model.

**Response:** Thank for reviewer's suggestions. The correlation analysis of observed NO$_3^-$ concentration and independent variables were performed to examine the possible factors for the spatiotemporal variability of NO$_3^-$ concentrations. It is only a tentative procedure to develop the machine-learning model. Due to coarse temporal resolution of socioeconomic data (e.g., GDP), we have to use some methods to resample these data to the month resolution. First of all, population density (PD) and GDP in 1995, 2000, 2005, 2010, and 2015 were linearly interpolated to calculate PD and GDP in each year during 2005-2015. Then, the yearly GDP data were uniformly divided by 12 to estimate the monthly GDP. The monthly PD equaled to the corresponding yearly PD. In our study, socioeconomic data were allocated to the month scale in a very simple way because the monthly GDP in each city were not available. The importance values of the ensemble model have been shown in Fig. S7. In the final model, all of the variables except GDP, PD, and grassland have been applied to estimate the ambient NO$_3^-$ concentrations across China. The regression coefficients including A, B, C, and the residual error (e$_{ij}$) determined by the MLR model were 0.42, 0.77, 0.09, and -0.87, respectively.

**Comment 4:** To validate the excellent prediction performance of the ensemble model, detailed information about the observed data in Figure 4 are suggested to be labeled, such as sampling site, month, year etc. Data from sites far from the selected training sites and covering key areas will be

more convincing and preferred.

**Response:** I agree with reviewer's suggestions. In our study, we used some unlearned data collected from previous references to validate the performance of the developed model. The detailed site information including latitude, longitude, year, and month have been shown in Table S2. Although most of these sites were located in East China, these sites were not close to the training sites and even several sites were located in Tibetan Plateau. Meanwhile, the observation data during 2006-2008 were also used to validate the temporal transferability of this model. Moreover, the site-based cross-validation was also applied to test the transferability of this model (Line 339-346). The overall result suggested the model (dataset) showed robust performance.

**Comment 5:** A comparison also to chemical transport model results such as GEOS-FP (http://wiki.seas.harvard.edu/geos-chem/index.php/GEOS-FP) modeled nitrate will be very valuable.

**Response:** Thank for reviewer's suggestions. We also want to compare the modelled nitrate in our study with the GEOS-FP nitrate product. Unfortunately, we only found global nitrate concentrations since December, 2017, which cannot be compared with the nitrate concentrations in 2005-2015 retrieved by our study. In the future work, we hope to estimate the recent nitrate concentration after 2017 and then compare with the GEOS-FP product when the updated observation data are available.

**#Reviewer 2:**

In this study, Rui et al., attempt to use an ensemble of models to provide spatially and temporally resolved particulate nitrate concentrations in China. Of course, the availability of a reliable nitrate estimation across China would have immense value, but I see a number of issues in the presentation of the material. The sub-models are all poorly described and frankly quite confusing. The authors should clarify that they understand these beyond simply clicking the necessary buttons in a software package. Moreover, use of the explanatory variables is muddled. We don't know which variables were used and which thrown out due to redundancy, or how consistently that was done across sub-models or time periods. Finally the MLR statistics that define the ensemble model are not provided, including no discussion of multicollinearity issues. Application of the ensemble model that followed is of course only relevant if the model itself is well established.

**Response:** Thank for reviewer's suggestions. We have significantly revised the manuscript based on reviewer's suggestions. The detailed description of the ensemble model has been added in the

revised version. Besides, some new methods have been added to validate the robustness of the ensemble model and the $NO_3^-$ dataset.

**Comment 1:** Based on Figure S1, the site in far northwest China only seems to have data from 2010-2011. This may dramatically impact the reliability of the method as that site is singularly located and likely unique in terms of the meteorological variable range, which the models are dependent on. Please comment on this data issue.

**Response:** Thank for reviewer's suggestions. Indeed, the site in Northwest China only has data during 2010-2011 because the long-term monitoring in the arid region is a hard task. To date, no long-term $NO_3^-$ dataset in Northwest China during 2012-2015. However, the lack of $NO_3^-$ dataset during this period did not significantly degrade the modelling performance of $NO_3^-$ estimates. In the revised version, we performed the site-based cross validation to demonstrate the spatial transferability of this ensemble model. The basic principle is that all of the sites were evenly classified into ten clusters based on the geographical locations. Afterwards, nine of ten were used to train the model and then test the model based on the remained one. After ten round, all of the observed values versus estimate values was considered to be the final result to validate the spatial transferability of this model. Based on the site-based cross validation, we found that the ensemble model showed the better transferability in predicting the $NO_3^-$ concentrations. Thus, the estimated $NO_3^-$ concentration in Northwest China could be considered to be reliable though the measured $NO_3^-$ dataset in this region since 2012 was missing.

**Comment 2:** Please provide information in the SI material and comment in the text about the monthly sample size of useable $NO_2$ column data.

**Response:** Thank for reviewer's suggestions. A total of 1636 $NO_2$ column data during 2011-2015 was applied to develop the ensemble model to estimate the national $NO_3^-$ concentration. Afterwards, 1554236 useable $NO_2$ column data during 2005-2015 was applied to predict the gridded monthly $NO_3^-$ concentrations across China. The detailed information has been added in the revised version (Table S1).

**Comment 3:** Sec 2.1: (i) These are described as monthly samples. Please clarify that they are continuously collected over a month and not a single 24-h sample each month. I believe the former is true, correct? (ii) you should include the reference to Xu et al., (2019) which reports NNDMN data from 2010-2015, (iii) within the Xu et al., (2019), they specify that the $NO_3$ data is in units of

ugN/m^3 and I wonder if you corrected for the molecular weights in this regard when you reported as $NO_3$ concentrations? Your values seem similar to the ugN/m^3 values from Xu et al., (2019); (iv) finally, please specify the PM size-cutoff. This will impact whether nitrate from soil dust influenced the measurements. Xu, W., Zhang, L. & Liu, X. A database of atmospheric nitrogen concentration and deposition from the nationwide monitoring network in China. Sci Data 6, 51 (2019). https://doi.org/10.1038/s41597-019-0061-2

**Response:** Thank for reviewer's suggestions. We have added the description about the samples were continuously collected over a month in the revised version (Line 138). Besides, the reference of Xu et al. (2019) has been cited in the paper. The unit of $\mu g/m^3$ has been changed into $\mu g\ N\ m^{-3}$. The PM size-cutoff of $NO_3^-$ particles collected from the nationwide monitoring network was $PM_{10}$ (Line 136).

**Comment 4:** Sec. 2.3: with all of the variables introduced here, were they all incorporated into the MLR or just the sub-models? In the last paragraph of the section you mention, they are all incorporated into the model, but you don't indicate which model(s). Please clarify. And please provide detailed information about the finalized variables and which were found to be redundant. And was this made to be consistent for each time period? Or the final predictor variables varied with time period analyzed?

**Response:** Thank for reviewer's suggestions. The independent variables were incorporated into the sub-models rather than MLR. The simulated $NO_3^-$ concentrations for three sub-models were then input into the MLR to further estimate the $NO_3^-$ concentrations based on the ground-level $NO_3^-$ concentrations. The method is a two-stage model, which could enhance the modelling performance of $NO_3^-$ estimates. At first, all of the variables were input into the machine-learning model and then to remove some redundant predictors.

All of the independent variables were input into the sub-model rather than MLR. The simulated $NO_3^-$ levels from RF, GBDT, and XGBoost were integrated into the MLR model, which was the second-stage procedure.

Finally, all of the variables except GDP, PD, and grassland have been applied to estimate the ambient $NO_3^-$ concentrations across China.

In our study, we did not consider the time variability of the importance of independent variables for the $NO_3^-$ estimates because the training sample size was not large. However, the time (month of

year) was incorporated into the training model as a key variable because the time autocorrelation of $NO_3^-$ levels in the adjacent month might be paid more attention. As shown in Fig. S7, the time autocorrelation was not the most important variable compared with $NO_2$ column, and thus we did not need to analyze the time variability of the importance of independent variables for the $NO_3^-$ estimates.

**Comment 5:** Sec. 3.1 Equations 1-5: I (capital i), L, and chi are all not defined and generally make the use of this model too murky to understand

**Response:** Thank for reviewer's suggestions. where $(x_i, y_i)$ denotes the sample for $i = 1, 2, …, N$ in M regions $(M_1, M_2, …, M_z)$; $I$ denotes the weight of each branch; L denotes the branch of decision tree; $c_m$ represents the response to the model; $\overset{\Delta}{c_z}$ denotes the best value, m represents the feature variable; $c_1$ denotes the mean value of left branch; $c_2$ denotes the mean value of right branch; n is the split point. The detailed revision is shown in the body text.

**Comment 6:** Equation 6: R is not defined; c would be better to have a subscript, perhaps naught?; what is "arg"?

**Response:** Thank for reviewer's suggestions. $R_{tj}$ denotes each leaf node for the decision trees. c is regarded as the optimal value when $c_{tj}$ reaches the least value, which is not a naught. "argmin" denotes that the least value of the equation of $\sum_{xi \in Rt_j} L(y_i, f_{t-1}(x_i) + c)$, which equals to minimum.

**Comment 7:** Equation 8: again, undefined symbols and a general lack of description do not give the reader much confidence that the authors understand the complexities of the models that they are using

**Response:** Thank for reviewer's suggestions. It is a negligence in our study. All of the symbols were defined in the equation 8 in the revised version.

**Comment 8:** Sec. 4.2: I don't see the MLR statistics anywhere. Coefficients? Standard error of the coefficients? The MLR should also indicate whether or not it was really helpful to stack the models in the first place. You would expect this to suffer from multi-collinearity problems in that all three sub-models are attempting to predict the same nitrate concentrations. The authors can provide a variance inflation factor with interpretation in the text.

**Response:** Thank for reviewer's suggestions. The regression coefficients including A, B, C, and the residual error $(e_{ij})$ determined by the MLR model were 0.42, 0.77, 0.09, and -0.87, respectively. The

standard error of A, B, and C were 0.02, 0.03, and 0.01, respectively. We employed the stacking model of three machine-learning models because they could achieve the better performance compared with the individual model or the two-stage models. As shown in the following figure, we found that the $R^2$ values of RF+GBDT, GBDT+XGBoost, and RF+XGBoost models were lower than the stacking model of three machine-learning algorithms (see the following figure). Furthermore, both of RMSE and MAE for two-stage models were higher than those of stacking model of three machine-learning algorithms. Based on the result, the stacking model of RF, GBDT, and XGBoost were applied to estimate the ambient $NO_3^-$ concentrations across China. Furthermore, the MLR model did not suffered from significant multi-collinearity problems. The variance inflation factors of RF (2.01), GBDT (2.69), and XGBoost (2.08) were significantly lower than 10, which suggested the MLR model was robust.

[Figure]

**Comment 9:** Figure 2: shouldn't each of the sub-models be connected with the original data tree ($NO_2$ column / met data / land use) to show their optimization before incorporation into the MLR?

**Response:** Thank for reviewer's suggestions. All of the hyperparameters were adapted to ensure the optimization of each sub-model, and then the simulated $NO_3^-$ concentrations by each sub-model was integrated into the MLR model to further estimate the $NO_3^-$ concentration. The ensemble model is

two-stage model, and MLR model is an optimized process. All of the independent variables including $NO_2$ column, meteorological conditions were only incorporated into the sub-model instead of MLR model. In the MLR model, only the simulated values by RF, GBDT, and XGBoost were input.

**Comment 10:** Figure 3: (i) something is missing from the caption. You seem to have forgotten to label what is 3a or 3d, (ii) please include sample size for each plot; (iii) why is there a different sample size for each plot?

**Response:** Thank for reviewer's suggestions. We have redrawn the figure 3 in the revised version and the caption has been significantly revised. The sample size of each plot is 1636. Besides, the sample size of each plot has been shown in the same colorbar.

**Comment 11:** Figure 4: (i) I don't understand why there is a color bar...is it for the number of points reported from another study? how can it be between 1 and 2?, (ii) the boxes are different sizes. Does it mean that these represent distributions of predicted $NO_3$ concentrations? How to interpret the regression in this manner?

**Response:** Thank for reviewer's suggestions. Yes, the colorbar denotes the number of points. Figure 4 reflects the correlation of observed $NO_3^-$ concentrations collected from previous studies and corresponding simulated ones in the same grid and during the same period. These data were collected from previous studies to validate the transferability of the ensemble model because these data obtained from previous studies were not used to develop the model, which can be treated as the unlearned dataset. We have redrawn Figure 4 and ensured the same size of these points. The regression curve in this figure represents the optimal fitting curve of estimated $NO_3^-$ levels and observed ones.

**Comment 12:** Figure 5: In the caption, Please specify that these values are from your ensemble model output

**Response:** I agree with reviewer's suggestions. I have rewritten the caption.

**Comment 13:** Figure 6: again, please specify these are from ensemble model output. "Satellite-derived" could be mistaken for reanalysis data

**Response:** I agree with reviewer's suggestions. I have rewritten the caption.

**Comment 14:** Figure 7: (i) are these averaging all grid cells within a certain boundary? Are these boundaries specified anywhere? Please provide details in the text (ii) add the average measured

trends for plots b,c, and d

**Response:** Thank for reviewer's suggestions. Indeed, the inter-annual $NO_3^-$ concentrations in China, BTH, YRD, and PRD were estimated based on the average values of $NO_3^-$ concentrations within a certain boundary. The detailed information has been added in Fig. S1. The ambient $NO_3^-$ level in BTH showed the remarkable increase during 2005-2013 by 0.20 μg/m³/year. Afterwards, the $NO_3^-$ level decreased rapidly from 2013 to 2015 at a rate of -0.58 μg/m³/year. The $NO_3^-$ concentrations in YRD (0.11 μg/m³/year) and PRD (0.05 μg/m³/year) both showed the slight increases during 2005-2013, though the statistical test revealed the increases were significant. However, the $NO_3^-$ concentrations in YRD and PRD showed the dramatic decreases with -0.48 and -0.36 μg/m³/year during 2013-2015, respectively.

**Comment 15:** Figure 8: plots c and f are confusing. For most of the map, you show a change of only -0.1 to 0 ug/m³ and with a significant trend!. Are these comparing two annual values each with n=12 at each grid cell? or 24 points in sequence? Even if the latter, it seems incredible that less than a 0.1 ug/m³ change tested significant over 24 points

**Response:** Thank for reviewer's suggestions. It is a big fault in our study. The colorbar in Fig. 8c is misinformed. We have corrected the errors in the revised version. For the significance test of $NO_3^-$ trends, we used Mann-Kendall method to perform the trend analysis of $NO_3^-$ concentration, which could test whether the $NO_3^-$ concentration suffered from the significant trend during some periods. The method is a very old statistical technique developed by Mann (1945) and Kendall (1975) and has been applied in many fields (e.g., hydrology, atmospheric environment, meteorology). We performed the analysis based on the simple equation summarized in these references.

**Comment 16:** Table 1: (i) should the first column be labeled "year"? or you intended to add more data to the table? (ii) somewhere you should indicate the sample size for the various years

**Response:** Thank for reviewer's suggestions. The "season" has been changed into "year". The sampling size has been added in Table 1-3.

**Comment 17:** Figure S1: plots should be labeled with year of data or with a letter and specified in the caption

**Response:** I agree with reviewer's suggestions. The year of data has been added in the plots (Fig. S2).

**Comment 18:** Figure S3: all abbreviations should be described in the figure caption

**Response:** I agree with reviewer's suggestions. The full name of all the variables were added in the revised version (Fig. S4).

Other comments:

**Comment 19:** Lines 76-79: this sentence is difficult to understand. Please revise

**Response:** I agree with reviewer's suggestions. The sentence has been changed into "these sparse ground-observed sites did not accurately reflect the high-resolution $NO_3^-$ pollution especially the regions far away from these sites because each station only possessed limited spatial representative and $NO_3^-$ concentration was often highly variable in space and time" (Line 78-80).

**Comment 20:** Lines 106-107: please expand this sentence to clarify your meaning or perhaps delete it

**Response:** I agree with reviewer's suggestions. The sentence has been deleted.

**Comment 21:** Lines 117-119: this sentence is an example where you should specify "particulate nitrate concentration" and there are numerous other places that you should indicate this to distinguish from a possible misinterpretation of rainwater or cloudwater nitrate, the former which is also measured at NNDMN sites

**Response:** Thank for reviewer's suggestions. The particulate nitrate concentration has been emphasized in some other places to avoid the misinterpretation.

**Comment 22:** Line 127: NNDMN is not defined

**Response:** Thank for reviewer's suggestions. The full name of NNDMN is nationwide nitrogen deposition monitoring network (Line 129-130).

**Comment 23:** Line 135: this reference should be specified 2018a or 2018b or both

**Response:** I agree with reviewer's suggestions. Xu et al. (2018a) and Xu et al. (2019) were added in the revised version.

**Comment 24:** Lines 151-154: this is not a complete sentence, please clarify your meaning

**Response:** Thank for reviewer's suggestions. The sentence has been changed into "Among all of the daily meteorological data in ECMWF website, 2-m temperature ($T_{2m}$), 2-m dewpoint temperature ($D_{2m}$), 10-m latitudinal wind component ($U_{10}$), 10-m meridional wind component ($V_{10}$), sunshine duration (Sund), surface pressure (Sp), boundary layer height (BLH), and total precipitation (Tp) were applied to estimate national $NO_3^-$ levels" (Line 157-160).

**Comment 25:** Line 206: "to determine the appropriate parameter" ..... does it mean a regression

coefficient? or this is the 'c' value from the various models?

**Response:** Thank for reviewer's suggestions. These parameters denote the hyperparameters of each decision tree model. The maximum depth, minimum leaf node size, number of parameters for split, and number of trees of RF model was 8, 25, 6, and 500, respectively. The maximum depth, min_samples_split, and subsample ratio reached 7, 300, 0.7, respectively. The maximum depth, minimum child weight, and subsample ratio of XGBoost was 9, 5, and 0.7, respectively.

**Comment 26:** Lines 231-233: what does the first part of this sentence mean? "we only estimated the missing ratio of $NO_2$ column..." please clarify

**Response:** Thank for reviewer's suggestions. Satellite-based $NO_2$ columns generally suffered from some missing values, which were difficult to fill because the dataset of ground-observed $NO_2$ columns was not open access. These missing values might increase the uncertainty of $NO_3^-$ estimates.

**Comment 27:** Lines 242-243: specify these as measured monthly $NO_3^-$ concentrations

**Response:** I agree with reviewer's suggestions. The "ground-observed $NO_3^-$ concentrations" has been added in the revised version (Line 257).

**Comment 28:** Line 249: I don't see urban land area designations in Fig S2

**Response:** Thank for reviewer's suggestions. The "urban land area" has been deleted.

**Comment 29:** Lines 252-253: is there anything else to comment about this? any interpretation?

**Response:** Thank for reviewer's suggestions. The correlation analysis here is only to analyze the possible relationship between independent variables and $NO_3^-$ level, which is an exploratory analysis.

**Comment 30:** Lines 259-260: I don't understand why you say 'opposite trends' ....doesn't seem opposite. XGBoost is worst and ensemble best in all statistics

**Response:** Thank for reviewer's suggestions. The $R^2$ value showed the order of ensemble model > GBDT > RF > XGBoost, while the RMSE and MAE values followed the order of XGBoost > RF > GBDT > ensemble model. Thus, we believed that the $R^2$ value showed the opposite order with RMSE and MAE.

**Comment 31:** Line 269: "Furthermore," should be "However,"

**Response:** I agree with reviewer's suggestions. "Furthermore" has been replaced by "However" (Line 291).

**Comment 32:** Lines 277-279: I don't understand the latter half of this sentence. Please revise the

text

**Response:** Thank for reviewer's suggestions. The sentence has been changed into "The higher RMSE and MAE observed in 2010 might be contributed by the relatively scarce training samples, while the higher RMSE and MAE likely attained to the higher $NO_3^-$ levels during other years."

In fact, both of the relatively poor performance and high $NO_3^-$ levels caused the higher RMSE and MAE values. The sentence means the higher RMSE and MAE in 2010 was attributable to few training samples, while those in other years might be attributable to the higher $NO_3^-$ levels (Line 300-302).

**Comment 33:** Lines 280-281: is this for all years data combined? please specify

**Response:** Thank for reviewer's suggestions. Indeed, the seasonal performance was conducted based on all years data.

**Comment 34:** Lines 292-294: this sentence should start with "Although"

**Response:** Thank for reviewer's suggestions. The sentence starts with "Although".

**Comment 35:** Lines 303-304: I don't understand this sentence. Please revise the text

**Response:** Thank for reviewer's suggestions. The sentence has been changed into "At first, the predictive performances of Southwest China and Northwest China were significantly worse than that of NCP, thereby leading to the higher RMSE and MAE" (Line 326-327).

**Comment 36:** Lines 304-306 and 307-308: Does this explain why RMSE and MAE are poor in these areas? I don't see the definite connection

**Response:** Thank for reviewer's suggestions. The sample size is a very important factor for the accuracy of $NO_3^-$ estimates. Both of Northeast China and Northwest China possessed limited training samples (< 200), while the predictive performance of $NO_3^-$ estimates in Northwest China was significantly better than those in Northeast China. It was assumed that the sampling sites in Northeast China were very centralized, while the sampling sites in Northwest China were uniformly distributed across the whole region. Some previous studies have confirmed that the spatial distribution of monitoring sites significantly affected the $NO_3^-$ estimates for machine-learning models. In general, the uniform distribution of sites was beneficial to elevate the modelling accuracy of machine-learning models compared with the centralized distribution.

**Comment 37:** Lines 306-307: are these areas in Northwest China? please clarify

**Response:** Thank for reviewer's suggestions. Yes, these sites were located in Northwest China.

**Comment 38:** Lines 330-331: "especially for hindcast of air pollutants" .... you mean compared to the forecast of air pollutants? Seems strange that a forecast model would have a better statistical result

**Response:** Thank for reviewer's suggestions. We did not want to compare with the forecast result in this study. Thus, "hindcast" has been changed into "estimates" (Line 361).

**Comment 39:** Lines 365-368: the use of "speed" in these sentences is inappropriate. Example sentence revision for one of them: "For instance, the ambient $NO_3^-$ level in BTH increased remarkably by 0.13 $\mu g/m^3/year$ from 2005-2014."

**Response:** Thank for reviewer's suggestions. We have significantly revised the similar sentences in this paragraph based on the reviewer's suggestions (section 4.4).

**Comment 40:** Lines 391-400: decrease of $SO_2$ emissions but not $NO_x$ emissions can further lead to $NO_3^-$ increases because of decreased aerosol acidity, which is dictated by $SO_4^{2-}$ in particulate matter.

**Response:** I agree with reviewer's suggestions. The explanation has been added in the revised version (Line 439-441).

**Comment 41:** Line 461: does this mean average available $NO_2$ columns data was only ~43% each month? Or this was a minimum value and only at one site?

**Response:** Thank for reviewer's suggestions. The sentence means that number of available daily (not monthly) $NO_2$ columns across China during 2005-2015 accounts for about 43% of all the day-grids ($365 \times 11 \times 16320$), which included the $NO_2$ columns in all of the grids (0.25°) rather than those in the sites.

**#SC Reviewer:** In this study, the authors provided an ensemble model by stacking RF, GBDT, and XGBoost to acquire monthly ambient nitrate concentrations over China. Generally, the topic of this study is very interesting since national-scale products of ambient chemical components are of great importance. However, the adoption of datasets in this paper is not convincing.

**Response:** Thank for reviewer's suggestions. We have significantly revised the manuscript based on reviewer's suggestions. Meanwhile, many validation method has been applied to confirm the robustness of the ensemble model and the reliability of our dataset.

**Comment 1:** To be specific, the spatial distribution of ground sites (only 32) is very sparse, which means that they do not cover most of the study area. How could the authors ensure the accuracy of the whole study area using these ground truths? I wonder how to validate the result in the regions

without ground measurements, such as Tibet. Such regions are numerous in this study.

**Response:** Thank for reviewer's suggestions. In our study, some methods have been applied to confirm the robustness of this model. First of all, some unlearned data collected from previous references have been used to test the transferability of this model (Fig. 4). The result suggested that the $R^2$ value based on unlearned data was even higher than that of the developed model, indicating high accuracy of this dataset. This method might show some limitations due to scarce testing samples. Thus, we also employed a site-based cross validation method to examine the transferability of the ensemble model. The basic principle is that all of the sites were evenly classified into ten clusters based on the geographical locations. Afterwards, nine of ten were used to train the model and then test the model based on the remained one. After ten round, all of the observed values versus estimate values was considered to be the final result to validate the spatial transferability of this model. As depicted in Fig. S6, the site-based cross-validation $R^2$ value reached 0.73, which was slightly lower than the cross-validation $R^2$ value of the training model (0.78).

Indeed, some regions such as Tibet lacks of monitoring sites due to the difficulty of long-term observation, we cannot confirm the accuracy of this dataset directly. However, the site-based cross-validation and the validation based on some unlearned data have suggested that the dataset is reliable. In the future work, we will train a new model to further enhance the accuracy of $NO_3^-$ estimates when more samples are available.

**Comment 2:** Besides, GEOS-FP (http://wiki.seas.harvard.edu/geos-chem/index.php/GEOS-FP) can provide global 3-hour ambient nitrate concentrations at a similar spatial resolution. What is the main contribution of this study compared to GEOS FP? The authors need to justify the above issues in detail.

**Response:** Thank for reviewer's suggestions. We have tried our best to search the global 3-hour nitrate product in GEOS-FP website. Unfortunately, we only found the 3-hour nitrate concentrations since December, 2017. However, the long-term ambient nitrate concentrations during 2005-2015 were not available. The major contribution of our study is to obtain a long-term ambient nitrate concentrations across China and share with these data at an open website. Moreover, the data quality is convincing because the ground-level observation data has been assimilated into the model and the final model showed the robust performance. By contrast, the GEOS-FP product did not assimilate the ground-level $NO_3^-$ concentrations in China, which might significantly biased from

the real situation in China due to the uncertainty of emission inventory and imperfect mechanisms of chemical transport model

**Comment 3:** Some minor comments are listed below. Section 3: Why did the authors select these three machine learning methods for stacking? What if the authors only chose two of them?

**Response:** Thank for reviewer's suggestions. We employed the stacking model of three machine-learning models because they could achieve the better performance compared with the individual model or the two-stage models. As shown in the following figure, we found that the $R^2$ values of RF+GBDT, GBDT+XGBoost, and RF+XGBoost models were lower than the stacking model of three machine-learning algorithms. Furthermore, both of RMSE and MAE for two-stage models were higher than those of stacking model of three machine-learning algorithms. Based on the result, the stacking model of RF, GBDT, and XGBoost were applied to estimate the ambient $NO_3^-$ concentrations across China.

[Figure]

**Comment 4:** Fig. 2: I notice that this flowchart is very similar to those in the authors' previous publications (e.g., Developing a novel hybrid model for the estimation of surface 8h ozone ($O_3$) across the remote Tibetan Plateau during 2005-2018). Maybe a new style would be better.

**Response:** I agree with reviewer's suggestions. A new style workflow has been added in Figure 2.

**Comment 5:** Line 206: The parameters for RF, GBDT, and XGBoost are not given. Please provide

**Response:** Thank for reviewer's suggestions. The maximum depth, minimum leaf node size, number of parameters for split, and number of trees of RF model was 8, 25, 6, and 500, respectively. The maximum depth, min_samples_split, and subsample ratio reached 7, 300, 0.7, respectively. The maximum depth, minimum child weight, and subsample ratio of XGBoost was 9, 5, and 0.7, respectively.

**Comment 6:** Fig. 3: XGBoost shows the worst performance, which is unusual. The authors need to provide some discussions. Did this happen in other literatures?

**Response:** Thank for reviewer's suggestions. To the best of my knowledge, the performance of RF and XGBoost was strongly dependent on the training dataset and sampling size. Indeed, some previous studies have verified that XGBoost often showed the better performance compared with RF. It was assumed that XGBoost showed the better performance for big-data samples. However, the size of training samples (1636) in our study was relatively less than those in previous studies. Xiao et al. (2018) also verified that the XGBoost showed the better accuracy than RF in some developed regions such as East China, while RF showed the better performance than XGBoost in Northwest China because the monitoring sites in Northwest China was relatively scarce. We have added some discussions in the revised version (Line 279-285).

**Comment 7:** Fig. 5: Some point-shaped high values exist in the results (e.g, Northern China), which look like noises. Is this spatial distribution correct?

**Response:** Thank for reviewer's suggestions. Some point-shaped high values exist in North China because some urban areas were located in these areas. Besides, the high-resolution $NO_2$ columns could reflect the information of local industrial points. As shown in Fig. S7, urban area showed the higher importance, and thus some hot spots of $NO_3^-$ concentrations were observed in these points. Overall, the spatial distribution of $NO_3^-$ concentration was in good agreement with $NO_2$ column and surface $NO_2$ concentrations estimated by previous studies (Zhan et al., 2018 EST).

---

## Author Comment (AC3) · 27 Feb 2021

The comment was uploaded in the form of a supplement:
https://essd.copernicus.org/preprints/essd-2020-243/essd-2020-243-AC3-
supplement.pdf